# Destruction and reinstatement of coastal hypoxia in the South China Sea off the Pearl River Estuary

Yangyang Zhao[1,2], Khanittha Uthaipan[1], Zhongming Lu[3], Yan Li[1], Jing Liu[1], Hongbin Liu[4,5], Jianping Gan[3,5,6], Feifei Meng[1], Minhan Dai[1]

[1]State Key Laboratory of Marine Environmental Science, College of Ocean and Earth Sciences, Xiamen University, Xiamen, 361102, China
[2]Environmental Physics, Institute of Biogeochemistry and Pollutant Dynamics, ETH Zurich, Zurich, 8092, Switzerland
[3]Division of Environment and Sustainability, Hong Kong University of Science and Technology, Kowloon, Hong Kong SAR, China
[4]Division of Life Science, Hong Kong University of Science and Technology, Kowloon, Hong Kong SAR, China
[5]Department of Ocean Science, Hong Kong University of Science and Technology, Kowloon, Hong Kong SAR, China
[6]Department of Mathematics, Hong Kong University of Science and Technology, Kowloon, Hong Kong SAR, China

*Correspondence to*: Minhan Dai (mdai@xmu.edu.cn)

**Abstract.** We examined the evolution of intermittent hypoxia off the Pearl River Estuary during three cruise legs conducted in July 2018: one during severe hypoxic conditions before the passage of a typhoon and two post-typhoon legs showing destruction of the hypoxia and its reinstatement. The lowest ever recorded regional dissolved oxygen (DO) concentration of 3.5 μmol kg$^{-1}$ (~ 0.1 mg L$^{-1}$) was observed in bottom waters during Leg 1, with a ~ 660 km$^2$ area experiencing hypoxic conditions (DO < 63 μmol kg$^{-1}$). Hypoxia was completely destroyed by the typhoon passage but was quickly restored ~ 6 days later, resulting primarily from high biochemical oxygen consumption in bottom waters that averaged 14.6±4.8 μmol O$_2$ kg$^{-1}$ d$^{-1}$. The shoreward intrusion of offshore subsurface waters contributed to an additional 8.6±1.7 % of oxygen loss during the reinstatement of hypoxia. Freshwater inputs suppressed wind- and tidal-induced turbulent mixing, stabilizing the water column and facilitating the hypoxia formation. The rapid reinstatement of summer hypoxia has a comparable timescale with water residence time and that of its initial disturbance from frequent tropical cyclones throughout the wet season. This has important implications towards better understanding the intermittent nature of hypoxia and predicting coastal hypoxia in a changing climate.

## 1 Introduction

Coastal hypoxia has been increasingly exacerbated near the mouths of large rivers as a consequence of anthropogenic nutrient inputs (Gilbert et al., 2010; Rabalais et al., 2014; Breitburg et al., 2018). The rise in the size, intensity and frequency of eutrophication-induced hypoxia exposes coastal oceans to a higher risk of elevated N$_2$O and CH$_4$ production, enhanced ocean acidification and associated reductions in biodiversity, shifts in community structures and negative impacts on food security and livelihoods (Diaz and Rosenberg 2008; Vaquer-Sunyer and Duarte 2008; Naqvi et al., 2010).

Coastal hypoxia can be intermittent due to the dynamic nature of estuarine and coastal environments, where winds, tides, river discharge and circulation patterns strongly affect the ventilation of oxygen-deficient waters (Wang and Justić 2009; Lu et al., 2018; Zhang et al., 2019). Constraints on oxygen supply can be easily eroded by changes in physical forcings, leading to the temporal alleviation of hypoxia (Laurent and Fennel 2019). Despite the wide application of oxygen budget analysis and modelling to diagnose the dominant processes driving the formation and maintenance of hypoxia (Yu et al., 2015; Li et al., 2016; Lu et al., 2018), the evolution of intermittent hypoxia, such as the destruction and reinstatement of hypoxia from disturbance by tropical cyclones, remains to be better characterized (Testa et al., 2017). Specifically, the identification of key processes and timescale constraints for these hypoxia destruction and recovery processes is of critical importance in order to predict site-specific hypoxia and its cascading effects, and to forecast the long-term impact of hypoxia under a changing climate with higher-intensity extreme events (Knutson et al., 2010; Mendelsohn et al., 2012).

Large riverine nutrient loadings and the resulting eutrophication have recently tipped the lower Pearl River Estuary (PRE) and adjacent shelf areas into seasonally hypoxic systems (Yin et al., 2004; Rabouille et al., 2008; Su et al., 2017; Qian et al., 2018; Cui et al., 2019; Zhao et al., 2020). Modelling results have shown that summer hypoxia off the PRE is largely intermittent owing to high-frequency variations in wind forcing and tidal fluctuations (Wang et al., 2017; Huang et al., 2019). Hypoxia is often interrupted by the passage of typhoons, but redevelops quickly with a tendency toward rapid oxygen declines (Su et al., 2017; Huang et al., 2019). The prevailing southwest monsoon usually favours the expansion of a quasi-steady-state freshwater bulge outside the entrance of the PRE (Gan et al., 2009; Lu et al., 2018) that promotes water column stability. However, it remains unclear how the interaction between wind stress, tidal forcing and freshwater buoyancy affects the bottom oxygen conditions when the winds shift in the downwelling-favourable easterly or southeasterly direction, especially in the wake of tropical cyclones. Aerobic respiration of organic matter is largely responsible for the oxygen depletion here (Su et al., 2017; Qian et al., 2018). Considering the oxygen consumption rate (OCR) has primarily been estimated based on incubations examining bacterial or community respiration (Su et al., 2017; Cui et al., 2019; Li et al., 2019), the actual magnitude of *in situ* OCR during the hypoxia formation process has rarely been reported at large-scales. The role of lateral advection (and/or upwelling) also remains to be better quantified (Zhang and Li 2010; Lu et al., 2018; Qian et al., 2018; Cui et al., 2019).

We investigated the destruction and reinstatement of summer hypoxia off the PRE to examine the effects of freshwater inputs, winds and tides on water column stability and the maintenance, destruction and formation of hypoxia. With the aid of a three-endmember mixing model, we partitioned physical- and biochemical-induced oxygen sinks and calculated the OCR and timescale for hypoxia regeneration after its destruction by a typhoon. The impacts of tropical cyclones on the evolution of seasonal hypoxia in river-dominated ocean margins is further discussed.

## 2 Materials and methods

### 2.1 Study area and cruise background

The shelf of northern South China Sea (NSCS) receives an average annual freshwater discharge of ~10,000 $m^3$ $s^{-1}$ originating from the Pearl River, the 17[th] largest river in the world (Cai et al., 2004; Dai et al., 2014). Nearly four fifths of freshwater discharge occurs during the wet season, typically from April to September (Dai et al., 2014). The riverine freshwater extends offshore to form a widespread plume over the shelf in summer (Gan et al., 2009; Cao et al., 2011; Chen et al., 2017), via eight outlets through three sub-estuaries (i.e., Lingdingyang, Modaomen and Huangmaohai; Fig. 1b). On the inner shelf, coastal upwelling interacts with the buoyant plume, propelled by the prevailing southwest monsoon and intensified along the eastward widened shelf (Gan et al., 2009; Chen et al., 2017). Climatologically, about 7 tropical cyclones per year impacted the NSCS from 1949-2019, half of which featured maximum wind speeds greater than 32.7 m $s^{-1}$.

Field observations and sampling were conducted onboard the R/V Haike 68 off the PRE on the inner-shelf of the NSCS in the summer of 2018. The cruise was interrupted by the passage of typhoon SONTINH across the NSCS, ~ 350 km south of the PRE (Fig. 1a). Leg 1 (July 8-14) was the cruise period before the typhoon, and Leg 2 (July 21-25) and Leg 3 (July 26-29) were conducted after its passage (Fig. 1b). During each leg we collected samples from west to east and along the cross-shelf transects within isobaths of 10-35 m. Almost all stations in Leg 1 (56 stations) were revisited during Leg 2 (56 stations, including 4 stations differing from Leg 1), and nearly half again during Leg 3 (27 stations). Eight stations were additionally revisited on the way back to the port on July 31. Time-series observations with a sampling interval of 1 h were conducted at Station F303 for 26 h before Leg 2, beginning at 16:00 pm on July 19 (Fig. 1b). In contrast to the typical southwesterly winds in the NSCS with average monthly wind speeds of < 6 m $s^{-1}$ from June to September (Su 2004), easterly winds prevailed during the cruise period due to the typhoon, with the wind speeds increasing up to ~ 13 m $s^{-1}$ (Fig. 1c) at the Waglan Island to the east of the study area (Fig. 1b).

### 2.2 Sampling and analysis

Temperature and salinity were determined using a SBE 917 plus conductivity-temperature-depth recorder (SeaBird Electronics, Inc.). Discrete samples were collected using 5 L free-flow water samplers mounted onto a Rosette sampling assembly. Dissolved oxygen (DO), dissolved inorganic carbon (DIC), total alkalinity (TA) and Chlorophyll $a$ (Chl $a$) concentrations were measured at all stations with depth profiles from 1 m below the surface down to ~ 4-6 m above the bottom, generally at three depth layers. Additional high-resolution vertical samplings were conducted at 7-8 depth layers (Fig. 1b).

Salinity was calibrated against discrete water samples measured by a Multi 340i salimeter (WTW). The DO concentrations were measured onboard within ~ 12 h using the spectrophotometric Winkler method (Labasque et al., 2004), with a precision better than ±2 µmol $L^{-1}$. DIC was measured on ~ 0.5 mL acidified water samples using an infrared $CO_2$ detector (Apollo ASC-

3) with a precision of ±2 μmol L⁻¹ (Cai et al., 2004). TA was determined on 25 mL samples in an open-cell setting based on the Gran titration technique (see details in Cai et al.(2010)) with a Kloehn digital syringe pump. The analytical precision was ±2 μmol L⁻¹. Both DIC and TA concentrations were calibrated against certified reference materials provided by Dr. A. G. Dickson at the Scripps Institution of Oceanography, University of California, San Diego. Chl *a* concentrations were determined using a Trilogy laboratory fluorometer (Turner Designs, Inc.) after being extracted with 90% acetone for 14 h at -20 °C (Welschmeyer 1994) and calibrated using a Sigma Chl *a* standard.

## 2.3 Water column stability

Water column stability regulates the ventilation of subsurface waters and replenishment of DO by suppressing turbulent mixing with stratification (Obenour et al., 2012; Lu et al., 2018; Cui et al., 2019), which can be indicated by the buoyancy frequency (also known as the Brunt-Väisälä frequency),

$$N^2 = -(g/\rho)(\partial\rho/\partial z) \tag{1}$$

where $g$ is the gravitational acceleration, $\rho$ is potential density, and $z$ is the height above the seabed. Generally, a positive $N^2$ (i.e., $N^2 > 0$) indicates a stable regime where stratification may suppress turbulence (Tedford et al., 2009), and a larger $N^2$ value indicates a more stable water column.

## 2.4 Oxygen consumption rate

From the perspective of Euler observations and based on mass balance, the DO changes ($\Delta DO$) in subsurface waters over a specified time interval at a specific site can be decomposed into two components, one driven by physical mixing ($\Delta DO^{mix}$) and the other induced by biochemical processes ($\Delta DO^{bc}$). Here, we define the biochemical-induced DO consumption with time as the oxygen consumption rate (OCR). A higher OCR value indicates stronger oxygen consumption and a negative value indicates oxygen production via biochemical processes (e.g., photosynthesis). For revisited stations, $\Delta DO$ is the difference in DO values measured between the two sampling periods. The physical-mixing-induced DO variations were derived using a three-endmember mixing model, which construct the conservative mixing scheme among different water masses: brackish plume water (PW), offshore surface water (SW) and upwelled subsurface water (SUB) (Su et al., 2017; Cui et al., 2019; Zhao et al., 2020). The model is constrained by salinity (S) and potential temperature ($\theta$) according to the following equations:

$$f_{PW} + f_{SW} + f_{SUB} = 1 \tag{2}$$

$$S_{PW} \times f_{PW} + S_{SW} \times f_{SW} + S_{SUB} \times f_{SUB} = S^{meas} \tag{3}$$

$$\theta_{PW} \times f_{PW} + \theta_{SW} \times f_{SW} + \theta_{SUB} \times f_{SUB} = \theta^{meas} \tag{4}$$

where the superscript 'meas' denotes measured values, and $f$ represents the fraction that each endmember contributes to the *in situ* samples. Assuming that DO concentrations in surface waters before sinking to the depth were equilibrated with the atmosphere and the subsurface waters were isolated from the atmosphere due to restriction by stratification, these fractions were applied to predict conservative concentrations of DO ($DO^{mix}$) resulting solely from conservative mixing.

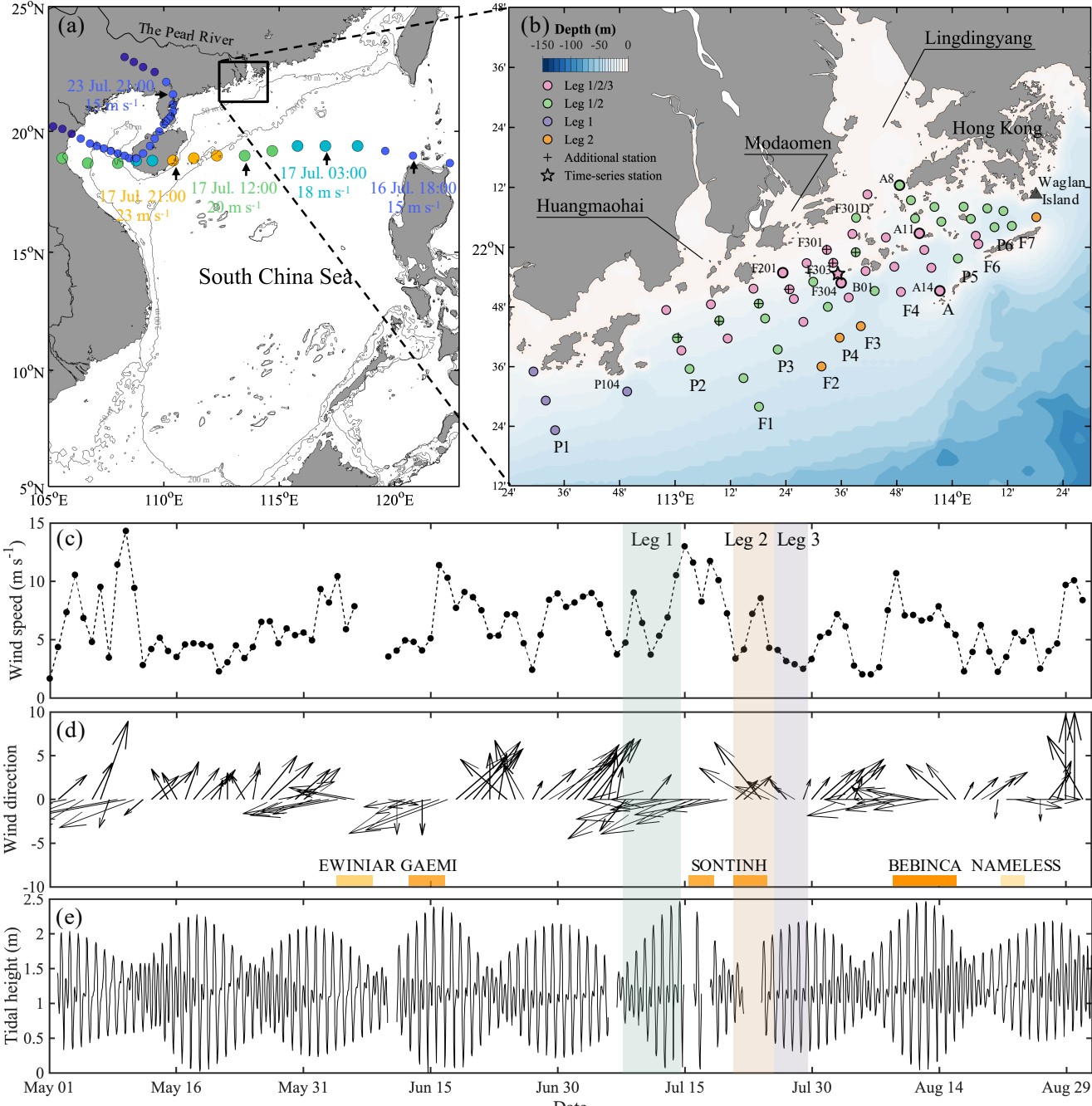

**Figure 1**: (a) Map of the study area on the shelf of the northern South China Sea (NSCS), showing the track of Typhoon SONTIHN (circles) across the NSCS during July 16-24, 2018. The color of the circles represents the magnitude of wind speed. Additionally, the smaller circles denote tropical depression (wind speeds ≤ 17.1 m s⁻¹) and the larger circles denote tropical storm (wind speeds within 17.2-32.6 m s⁻¹). The arrows denote the locations of the typhoon as marked with time and wind speed. The grey lines are the depth contours at 50 and 200 m. (b)

Sampling stations on the NSCS shelf off the Pearl River Estuary in summer 2018. The pink, green, purple and orange circles denote the stations surveyed in all three legs, only both Leg 1 and Leg 2, only Leg 1 and only Leg 2, respectively. Time-series observations were conducted at Station F303 as marked by the star, and vertically high-resolution samplings were conducted at stations marked with bold circles. (c) The wind speed and (d) wind direction at Waglan Island (triangle in (b)) from May to August, 2018. Bars at the bottom of (d) mark times when tropical cyclones impacted the NSCS. (e) The tidal height at the Dawanshan gauge station near Station F303 from May to August, 2018. The shaded area indicates the cruise periods for Leg 1 (grey), Leg 2 (pink) and Leg 3 (blue), respectively.

$$DO^{mix} = DO_{PW} \times f_{PW} + DO_{SW} \times f_{SW} + DO_{SUB} \times f_{SUB} \tag{5}$$

Similarly, $\Delta DO^{mix}$ is the difference in "conservative" DO values between visits, assuming that the bottom water masses where biochemical oxygen consumption prevailed were constrained by strong convergence and their outflow from the sampling area is insignificant on the time scale of the water residence time (Lu et al., 2018; Li et al., 2020). As a result,

$$OCR = - \Delta DO^{bc}/\Delta t = - (\Delta DO - \Delta DO^{mix})/\Delta t \tag{6}$$

where $\Delta t$ is the duration between the two observations at times $t_1$ and $t_2$ ($t_2 > t_1$), respectively, and $\Delta X = X^{t2} - X^{t1}$ ($X = DO$, $DO^{mix}$, $DO^{bc}$, etc.). The uncertainty in the calculation of OCR mainly derives from the estimation of conservative values predicted from the three-endmember mixing model. Sources of the composite uncertainty ($\varepsilon$) in derivation of $DO^{mix}$ are associated with potential temperature ($\theta$), salinity ($S$) and the dissolved oxygen (DO) values of endmembers.

$$\varepsilon_{DO^{mix}} = \sqrt{\sum_i^n \left[ \left( f_i \cdot \sigma_{DO_i} \right)^2 + \left( DO_i \cdot \sigma_{f_i} \right)^2 \right]} \tag{7}$$

where $\sigma_{DO}$ and $\sigma_f$ are uncertainties in the DO concentration and the fraction of each endmember $i$ (i.e., PW, SW and SUB), the latter of which can be calculated as

$$\sigma_{f_i} = \sqrt{\sum_j^n \left[ \left( \partial f_i / \partial \theta_j \cdot \sigma_{\theta_j} \right)^2 + \left( \partial f_i / \partial S_j \cdot \sigma_{S_j} \right)^2 \right]} \tag{8}$$

where $j$ also denotes each endmember.

## 2.5 Endmember selection and validations of the three-endmember mixing model

The potential temperature – salinity diagram is shown in Fig. 2a. The three-endmember mixing scheme for the bottom layer has been elucidated in Zhao et al. (2020) that had a spatial coverage similar to this study. We adopted the endmember values of the offshore surface water and upwelled subsurface water from Zhao et al. (2020) given that our sampling was almost exclusively within the 30-m isobaths (Fig. 1b) and these values were consistent with those found in previous studies (Cao et al., 2011; Guo and Wong 2015; Su et al., 2017). The brackish plume water was assumed to partly subduct to the bottom layer under downwelling favourable wind conditions (Huang et al., 2019; Li et al., 2021). The endmember values of the brackish plume water were thus determined from the surface water samples near the mouth of the PRE with a salinity of ~ 16.9, mainly consisting of a mixture of riverine freshwater and offshore surface water. The DIC endmember of brackish plume water here was consistent with the predicted value using the endmember values of riverine/plume water reported by Su et al. (2017), whereas it was higher than that calculated using the endmember values of riverine freshwater from Zhao et al. (2020) since the

riverine DIC concentrations might be diluted by abnormally high river discharge in 2017 (Guo et al., 2008). For simplification, DO concentrations in offshore surface water were assumed to be saturated, in equilibrium with the atmosphere, while the upwelled subsurface water was assumed to be oxygen-deficient by ~ 16% relative to the saturation level. The DO endmember value of brackish plume water was also assumed to be equilibrated with the atmosphere, which should be in order because the biological productivity was largely limited by high turbidity in shallow estuarine waters. A summary of the end-member values is listed in Table 1. In estimating the OCR, we excluded the above-pycnocline samples collected at depths < 10 m affected by the upper plume waters that are subject to strong air-sea exchanges and/or photosynthetic production of oxygen.

The predicted quasi-conservative TA ($TA^{pre} = TA_{PW} \times f_{RW} + TA_{SW} \times f_{SW} + TA_{SUB} \times f_{SUB}$; same for $DIC^{pre}$) is mostly consistent with our measured values (Fig. 2b), with a subtle difference of 8±8 μmol kg$^{-1}$ likely associated with measurement errors, computational errors in the mixing scheme and/or biological processes. The slope of ΔDIC ($\Delta DIC = DIC^{meas} - DIC^{pre}$) vs. ΔDO in bottom waters was -0.93±0.07 (Fig. 2c), similar to that reported by Zhao et al. (2020).

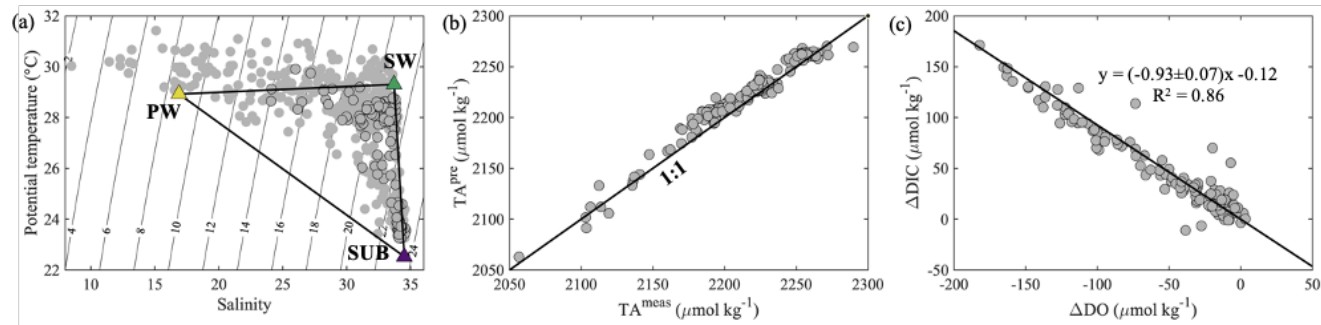

**Figure 2**: (a) Potential temperature (°C) vs. salinity, (b) predicted TA ($TA^{pre}$, μmol kg$^{-1}$) vs. measured TA ($TA^{meas}$, μmol kg$^{-1}$), and (c) ΔDIC (μmol kg$^{-1}$) vs. ΔDO (μmol kg$^{-1}$) on the NSCS shelf off the PRE. The black-edged circles represent bottom water samples with depths > 10 m. The yellow, green and purple triangles in (a) represent the endmember values of brackish plume water (PW), offshore surface water (SW) and upwelled subsurface water (SUB), respectively. The black line in (c) denotes the slope of ΔDIC plotted against ΔDO derived from the Model II regression.

**Table 1:** Summary of the end-member values adopted in the three-endmember mixing model

| Water mass | θ (°C) | Salinity | DIC (μmol kg$^{-1}$) | DO (μmol kg$^{-1}$) |
|---|---|---|---|---|
| Brackish plume water | 28.9±0.4[b] | 16.9 | 1776±29[b] | 217.3±1.4[c] |
| Offshore surface water[a] | 29.3±0.1 | 33.7±0.1 | 1922±5 | 194.4±0.3[c] |
| Upwelled subsurface water[a] | 22.5±0.1 | 34.5±0.0 | 2022±3 | 180.9 |

[a]Adopted from Zhao et al. (2020)
[b]Uncertainties were derived from multiple samples collected at the entrance of the PRE
[c]Uncertainties were calculated by propagating errors associated with the estimation of oxygen solubility using Benson and Krause (1984)

## 3 Evolution of intermittent hypoxia off the PRE

### 3.1 Extensive hypoxia before the typhoon

The water columns showed a prominent two-layer structure on the inner NSCS shelf off the PRE (Fig. 3 and Fig. 4). In the surface layer, the freshwater plume mainly attached the coast when veering to the west as constrained by estuarine topography, the Coriolis force (Wong et al., 2003), and easterly winds during Leg 1 (Fig. 1d), despite a freshwater bulge that remained near the mouth of the Lingdingyang sub-estuary due to persistent southwesterly winds before the cruise (Fig. 1d) and the weak shelf current there (Gan et al., 2009; Lu et al., 2018). A strong bloom occurred in the surface plume waters near the Huangmaohai sub-estuary, characterized by high Chl *a* concentrations of > 20 µg L$^{-1}$ and oversaturated DO of > 300 µmol kg$^{-1}$ (equivalent to a DO saturation level > 150 %) (Fig. 3g, j). The freshwater bulge also featured a relatively weak bloom, with Chl *a* concentrations of ~ 10 µg L$^{-1}$ and DO of ~ 250 µmol kg$^{-1}$ (equivalent to a DO saturation level of ~ 125 %), likely owing to the high nutrient concentrations, a favourable water residence time and an abundance of photosynthetically active radiation (Lu and Gan 2015). The surface water temperature was mostly > 29 °C. Exceptions occurred to the southwest of Hong Kong (Fig. 3a) where the air temperature was 2-3 °C lower during our visits (Fig. S1 in the Supplement).

At the bottom layer, low-temperature (< 26 °C), high-salinity (> 33) shelf benthic waters intruded onshore to the 10-20 m isobaths below the surface plume (Fig. 4a, d). An extensive hypoxic zone (DO < 63 µmol kg$^{-1}$) developed beneath the freshwater bulge and extended westwards along the 20-30 m isobaths to the region off the Modaomen sub-estuary (Fig. 4g). To the east, a relatively weak hypoxic centre occurred adjoining the Hong Kong waters. Additionally, a smaller-scale hypoxic zone appeared under the surface bloom near the Huangmaohai sub-estuary, a region which was also ever reported but not fully covered by survey measurements (Su et al., 2017; Cui et al., 2019). The general pattern of hypoxic zones was similar to that found in summers of 2014 and 2017 (Su et al., 2017; Zhao et al., 2020), yet with a slight offshore shift. The minimum oxygen level was 3.5 µmol kg$^{-1}$ (~ 0.1 mg L$^{-1}$) in bottom waters at Station F303, lower than the previously reported minimum (~ 7 µmol kg$^{-1}$ at Station F304, Su et al., (2017)). Within the surveyed region, the total area of the hypoxic zone reached ~ 660 km$^2$ and the oxygen-deficit zone (DO < 94 µmol kg$^{-1}$) occupied ~ 1470 km$^2$, larger than those in summer 2014 (> 280 km$^2$ for the hypoxic zone and > 800 km$^2$ for the oxygen-deficit zone) when we surveyed in a smaller area (Su et al., 2017). These findings might nevertheless indicate that summer hypoxia off the PRE has been increasingly exacerbated in recent years (Su et al., 2017; Qian et al., 2018; Zhao et al., 2020).

### 3.2 Destruction of hypoxia by the typhoon

The spatial patterns of temperature, salinity, DO and Chl *a* concentrations all changed drastically from disturbance by intensified easterly winds during the typhoon period. Strong winds drove the warm, low-salinity and oxygen-saturated surface waters to mix downward, increasing temperature and DO concentrations and decreasing salinity in bottom waters (Fig. 4). Time-series observations at Station F303 before Leg 2 showed a vertically well-mixed water column in the first half, as

reflected by homogeneous distributions of intermediate temperature (~ 28 °C), salinity (~ 32), and DO levels (~ 180 μmol kg$^{-1}$) (Fig. 5a-c). The less well-mixed water column in the second half of the time-series observations likely resulted from the weakened winds, showing an upward intrusion of bottom waters which were slightly warmer than surface waters which lost heat to the low-temperature atmosphere (Fig. S1). With subdued winds that shifted to southwesterly in the following two days (Fig. 1c, d), the offshore spreading of the river plume supressed the upward intrusion of slightly warm bottom waters and facilitated the restoration of a two-layer water column, as observed in the second half of the time-series observations (Fig. 5) and during Leg 2 (Fig. 3 and Fig. 4). Stronger blooms than that during Leg 1 were identified in the surface plume, widely spreading from the mouth of the Lingdingyang sub-estuary to near the Huangmaohai sub-estuary (Fig. 3k), potentially fuelled by nutrients mixed upward from the depth in addition to riverine inputs (Wang et al., 2017; Qiu et al., 2019). The maximum Chl *a* concentration was > 40 μg L$^{-1}$ off the Modaomen sub-estuary, accompanied by an extraordinarily high DO concentration of > 350 μmol kg$^{-1}$ (Fig. 3h, k). The surface water temperature was ~ 29.0±0.5 °C, higher than that in bottom waters by 0.8 °C during Leg 2 (Fig. 3b and Fig. 4b).

In bottom waters, however, hypoxia had been completely destroyed due to strong reaeration in the wake of the typhoon travelling across the NSCS, replaced by a homogenous spatial distribution of relatively high DO concentrations (~ 171±16 μmol kg$^{-1}$) during Leg 2 (Fig. 4h). The cross-shore gradients in temperature and salinity were also largely relaxed, with isotropically elevated temperatures up to ~ 28 °C and decreased salinity (Fig. 4b, e). The mid-depth distributions of temperature, salinity and DO concentrations showed similar patterns as at the bottom layer (Fig. S2). Although the water column remained relatively well-mixed under the surface layer, freshwater buoyancy and weakened winds facilitated the revitalization of density stratification and subsequent oxygen decline below the pycnocline. Indeed, the bottom water DO concentration at Station F303 decreased by ~ 18 μmol kg$^{-1}$ compared to that in the time-series observations and was lower than that at adjacent stations by ~ 9-22 μmol kg$^{-1}$ when revisited during Leg 2 on July 22 (Fig. S3).

### 3.3 Reinstatement of hypoxia after the typhoon

With the dying-out of the typhoon after its landfall to the west of the study area on July 23 (Fig. 1a), the wind speed decreased to < 5 m s$^{-1}$ on July 25 while the wind direction remained from the southeast before it shifted to southwesterly on July 29 (Fig. 1c, d). The DO concentrations in bottom waters were noticeably lower (139-164 μmol kg$^{-1}$ shallower than 20-m isobaths) starting around July 23 in the second half of Leg 2 (Fig. 1b and 4h). During Leg 3 from July 26-29, hypoxia was present again in bottom waters, due to favourable conditions for its formation (Fig. 3 and Fig. 4). The surface layer warmed to over 30 °C, increasing vertical thermal gradients and strengthening the stratification (Allahdadi and Li 2017). The freshwater bulge of lower salinity (< 15) advected offshore around the Modaomen sub-estuary, with the offshore migration of the bloom (Fig. 4f, l) likely driven by the interaction between the seaward buoyant current and northeastward shelf current (Pan et al., 2014; Li et al., 2020). The Chl *a* concentrations near the entrances of the three sub-estuaries remained relatively high (> 10 μg L$^{-1}$), and the DO concentrations remained at high levels of > 250 μmol kg$^{-1}$, ~ 20 % over the saturation levels (Fig. 3i, l).

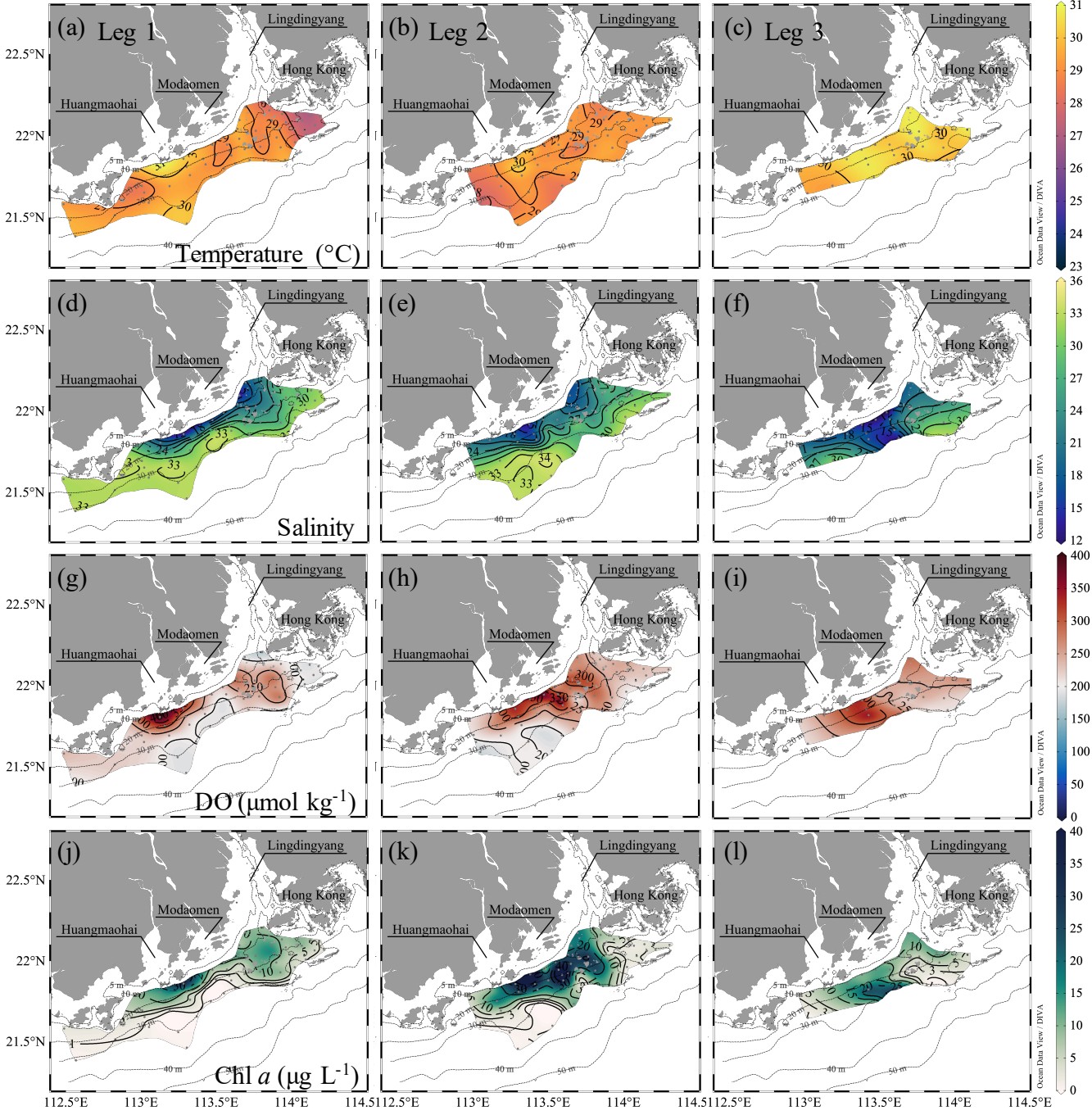

**Figure 3:** Distributions of temperature (°C), salinity, DO (µmol kg⁻¹) and Chl *a* concentrations (µg L⁻¹) at the surface water layer off the PRE during Leg 1 pre-typhoon, and during Legs 2 and 3 post-typhoon. The white and magenta contours in (g) and (w) show the hypoxic (DO < 63 µmol kg⁻¹) and oxygen-deficit (DO < 94 µmol kg⁻¹) zones. Figures were produced using Ocean Data View v. 5.3.0 (http://odv.awi.de, last access: 08 June 2020)

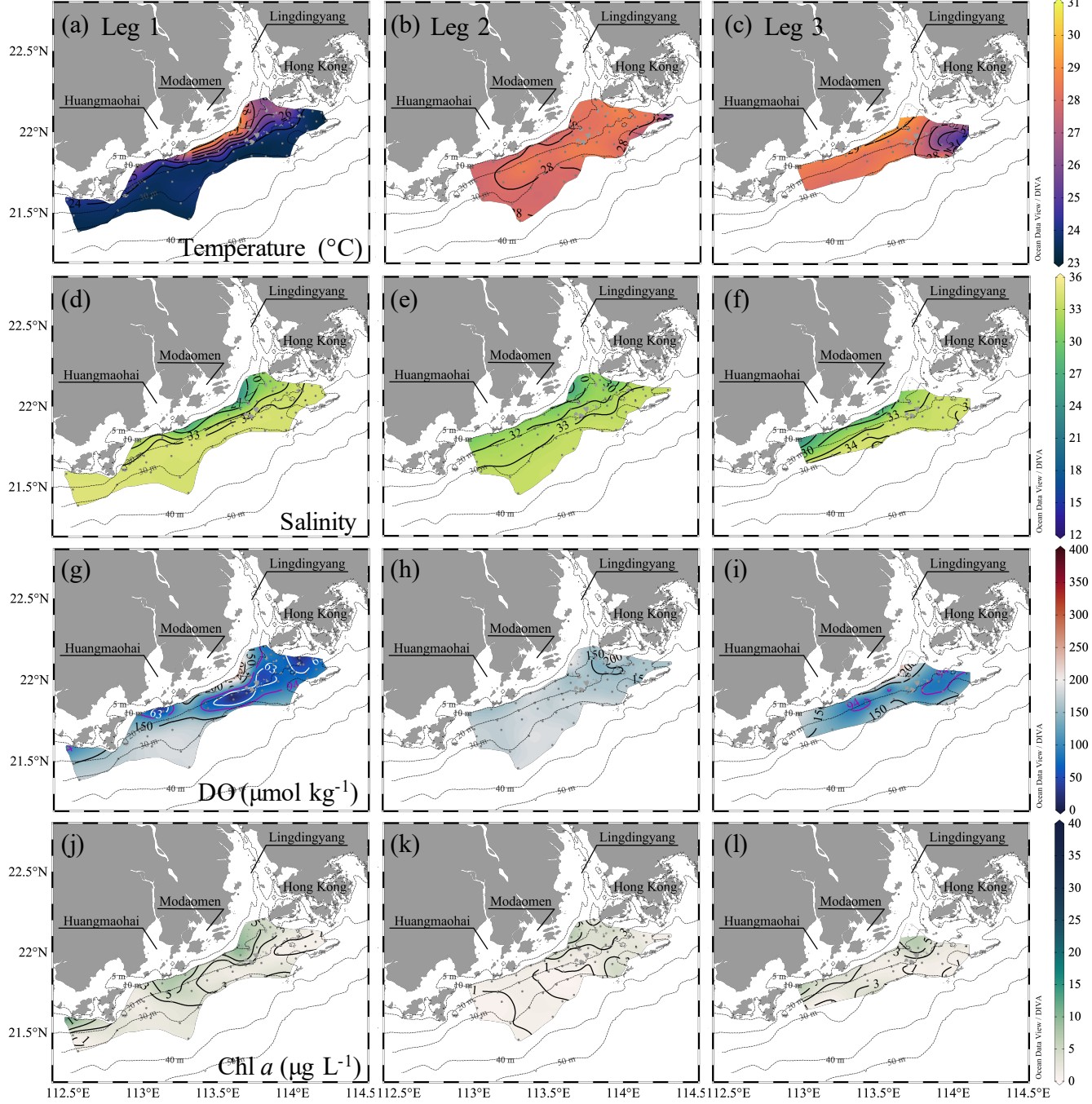

**Figure 4:** Distributions of temperature (°C), salinity, DO (μmol kg⁻¹) and Chl *a* concentrations (μg L⁻¹) at the bottom water layer off the PRE during Leg 1 pre-typhoon, and during Legs 2 and 3 post-typhoon. The white and magenta contours in (g) and (w) show the hypoxic (DO < 63 μmol kg⁻¹) and oxygen-deficit (DO < 94 μmol kg⁻¹) zones.

Similar to Leg 1 and Leg 2, the surface waters penetrated into the subsurface layer along the coast, likely forced by the downwelling-favourable winds (Huang et al. 2019, Li et al. 2021), augmenting temperature and DO concentrations but bringing down salinity, particularly in the mid-depth layer (Fig. S2). The downward penetration of surface waters, nonetheless, seemed to be restricted to the ~ 10-m isobath and thus offset only a limited amount of the oxygen reduction caused by biochemical consumption (Koweek et al. 2020). DO concentrations in bottom waters were reduced to ~ 46 μmol kg$^{-1}$ off the

Maodaomen sub-estuary along the 20-m isobaths and to < 94 μmol kg$^{-1}$ to the southwest of Hong Kong (Fig. 4i), indicating the reinstatement of hypoxia. When sites were revisited on July 31, the re-emerging hypoxia was found to have been strengthened, with oxygen levels down to ~ 37 μmol kg$^{-1}$, and expanded along the 20-m isobaths. We found that hypoxia formed in bottom waters with a temperature of ~ 28 °C off the Modaomen sub-estuary during Leg 3, while the oxygen-deficit zone to the southwest of Hong Kong showed a relatively low temperature of < 27 °C (Fig. 4c), likely due to the cross-isobath

transport of shelf benthic waters, arising from local topographic effects (Dai et al. 2014, Wang et al. 2014) as during Leg 1. In this sense, the shoreward intrusion of shelf benthic waters is not a prerequisite for the initiation of hypoxia formation off the PRE, but it still contributes to the reinstatement of hypoxia southwest off Hong Kong.

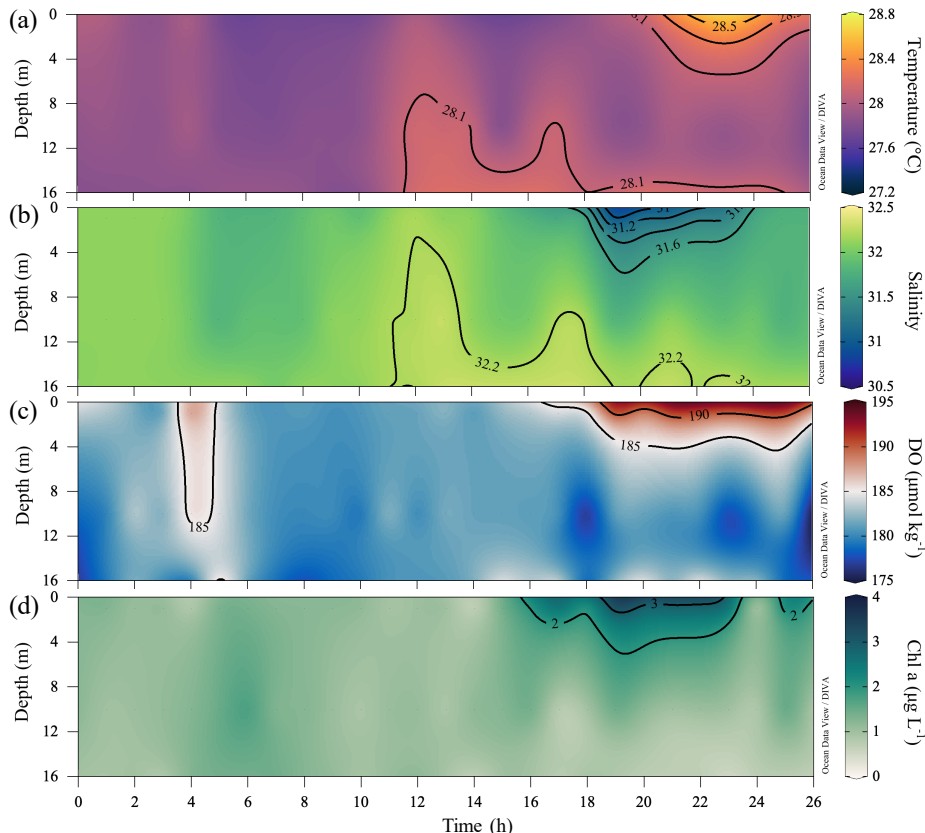

**Figure 5:** Time-series observations of (a) temperature (°C), (b) salinity, (c) DO (μmol kg$^{-1}$) and (d) Chl *a* concentrations (μg L$^{-1}$) at Station
F303 (see Fig. 1b) from July 19-20, 2018 after the typhoon passage, showing the complete destruction and the subsequent rapid development of stratification.

# 4 Maintenance, destruction and reinstatement of coastal hypoxia

## 4.1 Water column stability

A stable water column is a key prerequisite for the formation and maintenance of hypoxia in coastal oceans (Wang and Justić 2009; Obenour et al., 2012; Testa and Kemp 2014; Lu et al., 2018; Zhang et al., 2019), which restricts the oxygen supply by suppressing advective and diffusive mixing with oxygen-rich waters (Murphy et al., 2011; Cui et al., 2019). Many studies have demonstrated that density stratification becomes enhanced and stabilizes subsurface waters when freshwater flows over seawater (Gan et al., 2009; MacCready et al., 2009; Bianchi et al., 2010), allowing oxygen depletion over a longer timescale (Fennel and Testa 2019). In the second half of the time-series observations, the two-layer structure re-emerged with the spreading of the river plume, which suppressed the vertical mixing in the subsurface layer, as reflected by the receded upward intrusion of slightly warm, saline bottom waters (Fig. 5a, b). The Chl *a* concentrations in the surface layer increased, followed by an elevated DO level in surface waters and a lowered DO level in subsurface waters (Fig. 5c, d). The surface-to-bottom salinity difference showed large values within the surface plume area, which almost covered the bottom hypoxic zones (Fig. 4g, i). Exceptions only occurred to the hypoxic zone off the Modaomen sub-estuary, where the surface-to-bottom salinity differences were relatively small but the temperature differences were large (i.e., $\Delta T_{b-s} < -4\ °C$ in Fig. 6a-c) due to the shoreward intrusion of cold offshore subsurface waters (Fig. 4a-c). The regions occupied by the surface plume and the shoreward-intruded shelf bottom waters therefore overlapped, resulting in a more stable water column where a patchy hypoxic zone could persist for more than 5 days (Cui et al., 2019).

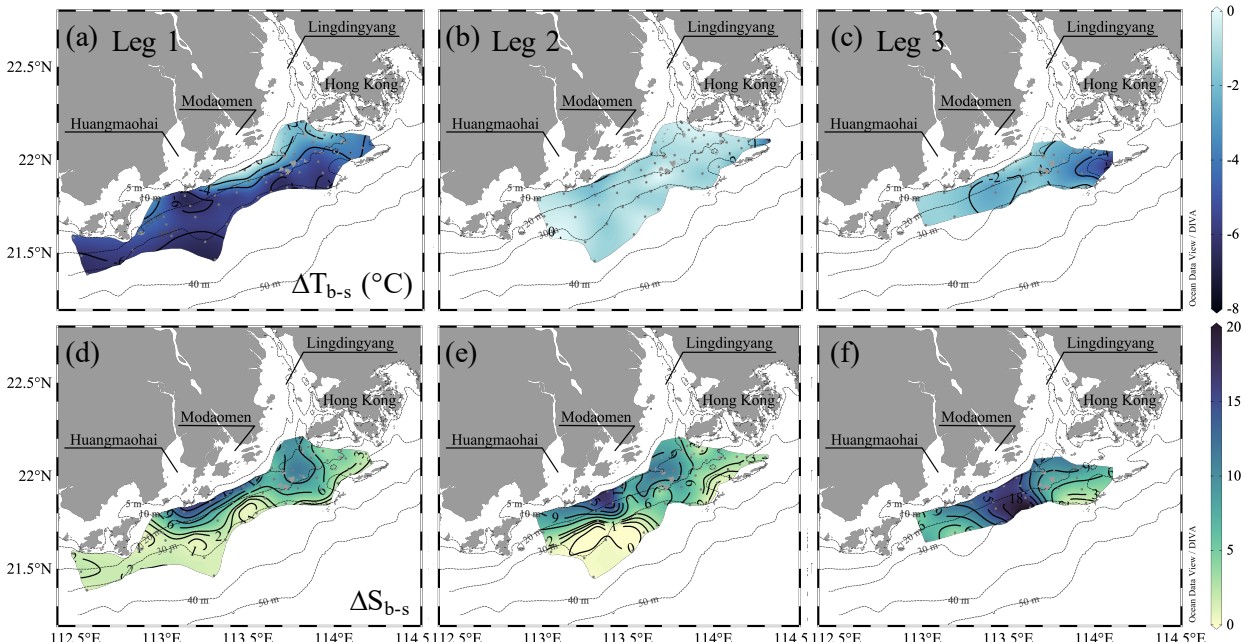

**Figure 6**: The surface-to-bottom temperature (a-c) and salinity (d-e) distributions off the PRE during Leg 1 pre-typhoon, and during Legs 2 and 3 post-typhoon. $\Delta T_{b-s}$ and $\Delta S_{b-s}$ represent the difference in temperature and salinity between the bottom and surface layer, respectively.

Off the PRE, when not influenced by freshwater inputs the surface layer showed a relatively small $N^2$ close to 0 and vertically well-mixed temperature, salinity and DO concentrations (e.g., in the top ~ 10 m at Station A14, Fig. 7c). However, in the presence of the freshwater plume, the surface layer became more stable, with a larger $N^2$ of > $1\times10^{-3}$, or even > $5\times10^{-3}$ (e.g.,
stations A8 and A11, Fig. 7a, b). The DO concentrations decreased sharply at the base of the surface plume (salinity ~ 30). Using the eddy diffusivity for density ($K_z$) of < $5\times10^{-6}$ m$^2$ s$^{-1}$ for $N^2$ larger than $1\times10^{-3}$ (Cui et al., 2019), we estimated the vertical diffusion for DO concentrations (VDIF = $K_z\times(\partial DO/\partial z)$) of ~ 0.25 g m$^{-2}$ d$^{-1}$ with a maximum of 0.54 g m$^{-2}$ d$^{-1}$ in the top 10 m at stations A8 and A11, which was comparable to the results from Cui et al. (2019). It therefore acted as a barrier layer, with weak dissipation of oxygen into the subsurface waters. The inherent pycnocline between the offshore surface water
and shelf benthic waters mainly driven by steep temperature gradients (Qu et al., 2007), such as at Station A14, acted as a second pycnocline in the plume region (e.g., stations A8, Fig. 7a), yet with weaker stratification likely from increased shear stresses in shallower waters (Pan and Gu 2016; Li et al., 2020). At the edge of the freshwater plume (Fig. 1b and Fig. 3a), such as Station F304, the second pycnocline was more stable than that induced by freshwater inputs (Fig. 7d). This three-layer structure, separated by two pycnoclines, therefore effectively decreased oxygen influx from the surface and facilitated oxygen
depletion in bottom waters.

Water column stability also largely depends on wind stress and/or direction in coastal waters (Wilson et al., 2008; Wang and Justić 2009). Higher wind stress usually de-stratifies the water column, leading to stronger turbulent mixing, air-sea gas exchange and reaeration (Chen et al., 2015; Lu et al., 2018; Huang et al., 2019), relieving hypoxic conditions (Ni et al., 2016;
Wei et al., 2016). During the typhoon period, the wind speed rose to as high as 13 m s$^{-1}$ (Fig. 1c), which was large enough to break the stratification (Geng et al., 2019) driven by freshwater inputs and the inherent thermocline. The strong winds facilitated mixing high-temperature, low-salinity surface waters and cold, saline bottom waters, resulting in a vertically-homogeneous temperature and salinity, as observed in the first half of the time-series observations before Leg 2 (Fig. 5). The sudden decrease in Chl $a$ concentrations in the surface layer might owe to dilution from the vertical mixing of surface plume
waters with subsurface seawaters (Qiu et al., 2019). The surface waters became undersaturated (~ 90 % of the oxygen saturation level), also likely due to the upward mixing of low-oxygen waters, which in turn favoured the ventilation of bottom waters and the breakdown of hypoxic conditions (Hu et al., 2017). Despite wind speeds still as high as 10 m s$^{-1}$ during Leg 2 (Fig. 1c), stratification was regenerated in the top ~ 10 m of the plume region, which had relatively large surface-to-bottom salinity differences (Fig. 6e) and high $N^2$ (Fig. 7e-h) under easterly winds (Fig. 1d). This suggests that freshwater input-induced
stratification suppressed turbulent mixing driven by wind stress, favouring the initiation of hypoxia development even under downwelling-favourable conditions. Most modelling works simulated the formation of hypoxia off the PRE under the typical summer southwest monsoon (Wei et al., 2016; Lu et al., 2018; Li et al., 2020; Yu et al., 2020); when southwest winds blow in a more southward direction, a larger hypoxic zone develops (Wei et al., 2016). In contrast to southwesterly winds that facilitate the wide eastward spreading of the surface plume (Gan et al., 2009), the downwelling-favourable easterly winds tend to
constrain the surface plume to flow westward near the coast (Li et al., 2021) as shown during Leg 2, and even drive the surface

oxygen-saturated waters to penetrate into the deep along the coast (Fig. S2 and Fig. 4), resulting in an offshore or westward shift of hypoxic zones with a limited spatial extent beneath the surface plume. If the easterly winds last for a longer time than the hypoxia formation timescale, stronger blooms in the surface plume (Fig. 3k) would enhance the bottom hypoxia with abundant supply of fresh, labile organic matters; but the downwelling-favourable winds also would destroy the bottom hypoxia
if the wind stresses become strong enough (Li et al., 2021).

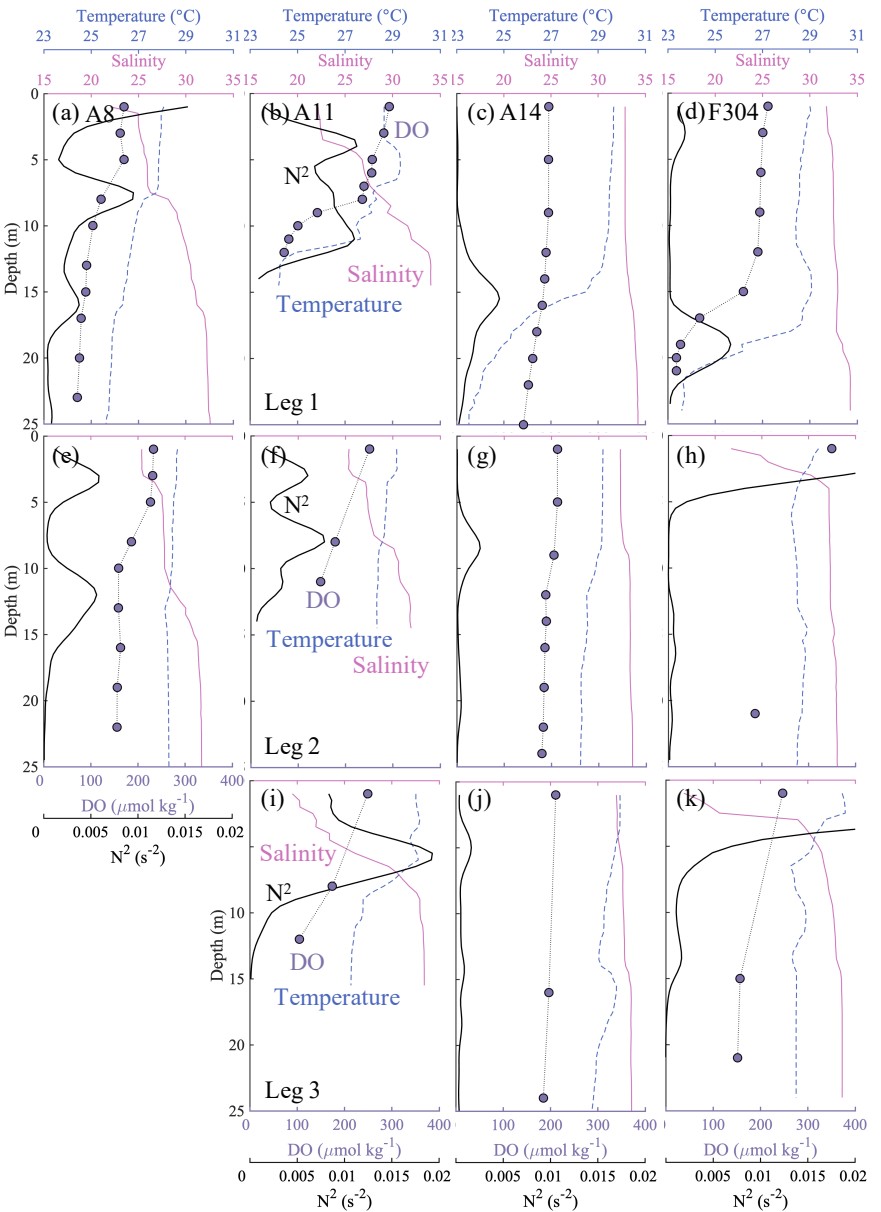

**Figure 7:** Profiles of temperature (°C) (green dashed lines), salinity (pink solid lines), dissolved oxygen (DO, μmol kg$^{-1}$) (purple dots) and buoyancy frequency $N^2$ (s$^{-2}$) (bold black solid lines) at stations A8, A11, A14 and F304 (see Fig. 1b), with visits both pre-typhoon (Leg 1) and post-typhoon (Legs 2 and 3). The vertical distributions of $N^2$ have been smoothed by the Gaussian method.

Tidal forcing has been suggested as another factor influencing the stability of the water column and the presence of hypoxia (Luo et al., 2009; Chen et al., 2015). Rabouille et al. (2008) demonstrated that tidal mixing could disrupt stratification and break hypoxia. Neap tides facilitate hypoxia formation relative to spring tides (Huang et al., 2019), and thus the intensity and area of hypoxia decreases during spring tides and increases during neap tides (Luo et al., 2009; Chen et al., 2015). As our observations were all conducted during the transformation from the neap tide to the spring tide (Fig. 1e), the intensity and spatial extent of the hypoxic zones off the PRE probably were underestimated. In the PRE, the dominant irregular semidiurnal mixed tide has a mean tidal range of 0.86-1.63 m, and a spring tidal range of 3.66 m (Mao et al., 2004). On the inner shelf, the spring tidal range can reach $\sim$ 2.5 m (Fig. 1e). The flood-ebb and spring-neap tidal oscillations lead to variations in the DO concentration outside of the Lingdingyang sub-estuary with a maximum neighbouring oxygen range of 0.2 and 0.5 mg L$^{-1}$, respectively (Cui et al., 2019). Assuming the observed DO concentrations in Leg 1 were overestimated by 0.5 mg L$^{-1}$ (i.e., $\sim$ 15 µmol kg$^{-1}$), the total area of the hypoxic zone and the oxygen-deficient zone would be at most $\sim$ 990 km$^2$ and $\sim$ 1930 km$^2$, respectively, 34-50% larger than our observed areas. The maintenance of hypoxia during Leg 1 and the reinstatement of hypoxic or oxygen-deficient conditions from Leg 2 to Leg 3 also suggested that the hypoxia off the PRE could survive from the spring tides under the conditions of a widespread plume and weak winds. Compared to freshwater inputs and winds, tidal forcing more likely acts as a secondary factor influencing the water column stability and the spatial extent of bottom hypoxia off the PRE (Luo et al., 2009; Chen et al., 2015; Zhang et al., 2019).

### 4.2 Oxygen sinks and hypoxia formation timescale

Oxygen sinks fundamentally drive the hypoxia formation under favourable physical conditions. Common methods to quantify oxygen sinks under hypoxic conditions include the analysis of oxygen budgets based on the mass balance of oxygen and estimates of community/bacterial respiration or nitrification rates using field incubations (Zhang and Li 2010; Li et al., 2015; Cui et al., 2019). However, the budget analysis of oxygen usually assumes a steady state system (Zhang and Li 2010; Cui et al., 2019), since the change of oxygen over time is much smaller than the oxygen depletion and advection/diffusion fluxes (Cui et al., 2019). The respiration or nitrification rates estimated from (enriched) incubation experiments also merely indicate the oxygen consumption rate (OCR) under a specific low-oxygen condition at the time of sampling (He et al., 2014; Su et al., 2017). It remains unclear about the magnitude of net oxygen sinks over time that actually leads to the oxygen decline for hypoxia formation.

On the condition of a precedent restoration of density stratification from Leg 2 to Leg 3, DO concentrations in bottom waters were generally reduced by > 25 µmol kg$^{-1}$, with two hotspots showing reductions up to 75 µmol kg$^{-1}$: one located offshore of the Lingdingyang sub-estuary, to the southwest of Hong Kong, and the other between the Modaomen and Huangmaohai sub-estuaries (Fig. 8a). The oxygen decline post-typhoon from Leg 2 to Leg 3 therefore provided a good case to partition physical- and biochemical-induced oxygen sinks and calculate the OCR and timescale for the reinstatement of bottom water hypoxia based on the three-endmember mixing model.

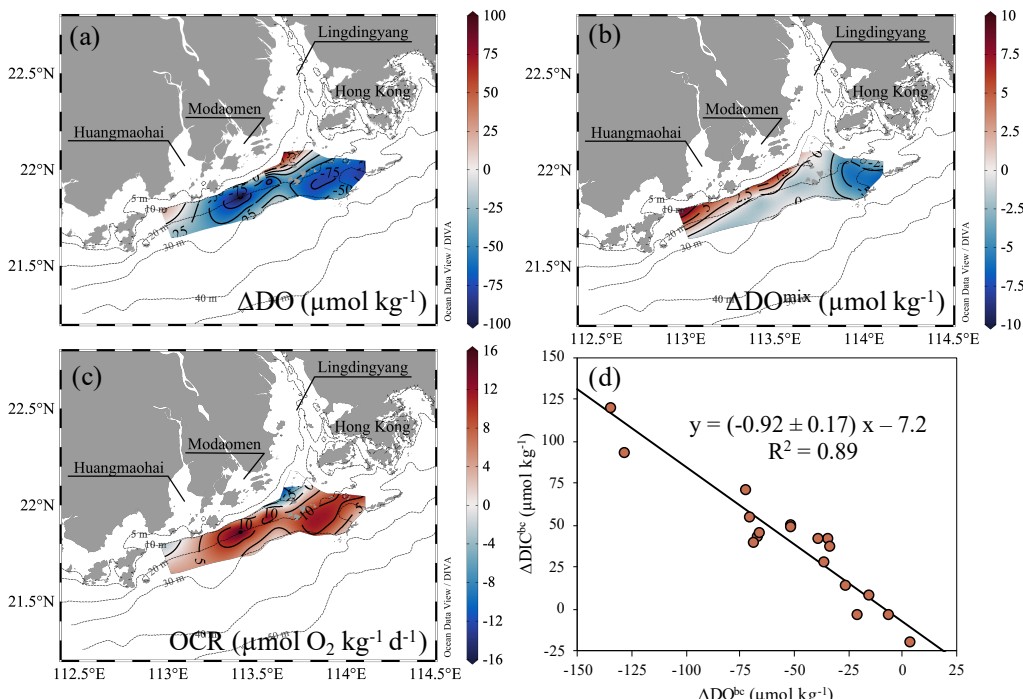

**Figure 8:** Distributions of (a) total DO changes (ΔDO, μmol kg$^{-1}$), (b) mixing-induced DO changes (ΔDO$^{mix}$, μmol kg$^{-1}$) and (c) the biochemical-induced oxygen consumption rate (OCR, μmol O$_2$ kg$^{-1}$ d$^{-1}$) between Leg 3 and Leg 2 on the inner NSCS shelf off the PRE. (d) The biochemical-induced changes in DIC (ΔDIC$^{bc}$, μmol kg$^{-1}$) vs. DO (ΔDO$^{bc}$, μmol kg$^{-1}$) in bottom waters with depths > 10 m from Leg 2 to Leg 3. The black line denotes the slope of ΔDIC$^{bc}$ plotted against ΔDO$^{bc}$ derived from the Model II regression.

### 4.2.1 Mixing-induced oxygen sinks

The mixing-induced DO changes were positive along the coast within 10-20 m isobaths, as the oxygen-saturated surface waters penetrated downward to re-aerate the bottom waters driven by the downwelling-favourable easterly winds (Huang et al., 2019). The mixing-induced oxygen sinks mainly occurred in bottom waters southwest off Hong Kong, with an average of -5.7±0.8 μmol kg$^{-1}$, higher than in other regions west of the PRE (e.g. beyond the 20-m isobath; -1.4±0.8 μmol kg$^{-1}$) (Fig. 8b). The mixing-induced oxygen sinks can be attributed to the shoreward intrusion of oceanic cold, oxygen-undersaturated subsurface waters — as reflected by lowered temperature (< 27 °C; Fig. 4c), which usually act as a non-local driver on coastal hypoxia by lowering the initial DO concentration (Wang 2009; Qian et al., 2017). The cold, saline oceanic subsurface waters also completely occupied the bottom layer beyond the 20-m isobath during Leg 1, where extensive hypoxia developed (Fig. 4a, g). This upwelling-induced reduction in the initial DO level amounted to 8.6±1.7 % of the oxygen decline to the southwest of Hong Kong, suggesting coastal upwelling played a minor role in the hypoxia formation.

The contribution of oxygen-deficient coastal upwelling to the hypoxia formation varies in different ocean marginal systems, which is largely dependent on the source of the subsurface water masses and biogeochemical reactions along the pathway of

the intrusion. Qian et al., (2017) showed the apparent oxygen utilization (AOU) values of ~ 50 µmol kg$^{-1}$ in the source waters of nearshore Kuroshio branch at the shelf-break northeast of Taiwan and an AOU increment of ~ 40 µmol kg$^{-1}$ throughout its travel time of ~ 60 days to the vicinity off the Changjiang River (Yang et al., 2013). In eastern boundary upwelling systems along the northeast Pacific Ocean, the source waters typically have the low DO concentration of ~ 80-160 µmol kg$^{-1}$, even near or below hypoxic levels from a depth of ~ 100-200 m (Grantham et al., 2004). The DO deficits were further exacerbated by respiration when the source waters transited shoreward over the shelf. Comparing to these systems, a higher DO level in the source waters (AOU ~ 35 µmol kg$^{-1}$) originating from the low-latitude, high-temperature oligotrophic NSCS (Wong et al., 2007) and a shorter shoreward travel time diverting their direction from the continental slope (Gan et al., 2009; Wang et al., 2014) might explain the relatively low contribution of coastal upwelling to the oxygen depletion on the inner NSCS shelf off the PRE.

### 4.2.2 Biochemical-induced oxygen sinks

The biochemical-induced DO and DIC changes from Leg 2 to Leg 3 also showed a good relationship with a slope of -0.92±0.17 (Fig. 8d), consistent with the slope of biochemical-induced variations in DO against DIC concentrations throughout the sampling legs (Fig. 2c), implying that aerobic respiration of organic matters indeed dominated the oxygen consumption. The distribution of OCR estimates in bottom waters almost mirrored variations in the total DO pattern from Leg 2 to Leg 3 (Fig. 8a, c). This biochemically-mediated OCR ranged from 0.9 µmol $O_2$ kg$^{-1}$ d$^{-1}$ at offshore non-hypoxic stations to 19.5±0.4 µmol $O_2$ kg$^{-1}$ d$^{-1}$ in hypoxic waters, with an average of 14.6±4.8 µmol $O_2$ kg$^{-1}$ d$^{-1}$ in the oxygen-deficit zone. The uncertainty introduced by the mixing scheme was 0.63-0.98 µmol $O_2$ kg$^{-1}$ d$^{-1}$, accounting for a deviation of 4-27 %. The spatial variability of the bottom OCR was not fully coupled with the location of surface blooms (Fig. 3l and Fig. 8c), even though eutrophication-produced organic matters were primarily responsible for fuelling oxygen depletion in the hypoxic zone (Su et al., 2017; Yu et al., 2020; Zhao et al., 2020). Cui et al., (2019) suggested that the patchy distribution of bottom hypoxia was closely associated with the river plume front which traps organic particles (Hetland and DiMarco 2008), accelerating their settlement and deposition in the overlapping zones between the river plume and shelf salt wedge (Zhang and Li 2010) to fuel the high OCR there. In fact, only the west hypoxic centre located beneath the surface bloom from Leg 2 to Leg 3 (Fig. 3j-l and Fig. 4g-i). Lu et al. (2018) proposed another explanation that small detritus of external sources could be accumulated and remineralized in the bottom flow convergence zone as regulated by highly variable coastline and bottom topography. Our observed hypoxic zones generally coincided with the modelled strong convergence zones (Lu et al., 2018; Li et al., 2020), implying sufficient organic matter supply fuelled the high OCR to restore hypoxia in bottom waters.

Our estimated OCR is comparable in magnitude to the community/bacterial respiration rate from previous studies in this study area (9.6 µmol $O_2$ kg$^{-1}$ d$^{-1}$, Su et al., (2017); 7.9 to 19.0 µmol $O_2$ kg$^{-1}$ d$^{-1}$, Cui et al., (2019); 16.8±8.9 µmol $O_2$ kg$^{-1}$ d$^{-1}$, Li et al., (2019)) and within the range in other estuaries and coastal systems (Dortch et al., 1994; Robinson 2008), despite a lower limit due to a potential overestimation of the actual time if significant oxygen consumption started later than our observations during

the first half of Leg 2 and an assumed negligible oxygen flux supplied from the surface by diffusion. From Leg 2 to Leg 3, the vertical oxygen diffusion flux was ~ 0.18 g m$^{-2}$ d$^{-1}$, amounting to a supply of ~ 0.6-1.1 μmol O$_2$ kg$^{-1}$ d$^{-1}$ to the bottom waters with a thickness of 5-10 m. The exclusion of this vertical oxygen diffusion flux in our estimates therefore might underestimate the OCR by ~ 6 % on average. We also assumed that the OCR was uniform in the subsurface waters several meters above the seabed because we collected samples at three depth layers during Leg 2 and Leg 3, with usually only one layer below the pycnocline for most stations. For stations with the middle layer also below the pycnocline, such as Station F304 during Leg 3, the profile of DO concentrations showed almost constant values below the pycnocline (Fig. 7k). Sediment oxygen demand might be significant near the seabed or in its overlying water column in shallow waters (Kemp et al., 1992; Zhang and Li 2010), but in our sampling area, oxygen losses by sediment oxygen demand (i.e., benthic respiration) were found much smaller than the bacterial respiration in the water column based on both incubation experiments and oxygen budget analysis (Cui et al., 2019). We thus assumed the sediment oxygen demand was negligible and the microbial respiration in the water column dominated the estimated OCR.

However, the estimated OCR here differs from that derived from the steady-state budget analysis in which the oxygen consumption was completely offset by the physical transport of oxygen (Zhang and Li 2010; Cui et al., 2019). During the hypoxia formation, the DO concentrations are in a non-steady state as oxygen sinks exceed sources. To shift towards a balance between oxygen consumption and replenishment for the maintenance of hypoxia, the OCR might decrease or the physical-induced oxygen supply increases. Kalvelage et al., (2015) found a significant correlation between OCR and *in situ* DO concentrations off Peru — aerobic respiration rates decreased at the upper boundary of oxygen minimum zone (OMZ) towards the OMZ core. The oxygen enriched incubation of unfiltered water samples also revealed that the OCR could be significantly enhanced when the initial *in situ* DO concentration was low (e.g., ~ 30 μmol kg$^{-1}$), but changed little when the *in situ* DO concentration was higher than ~ 90 μmol kg$^{-1}$ (He et al., 2014). Despite the bottom DO concentrations of ~ 180 μmol kg$^{-1}$ at Station F303 during the time-series observations, DO declined at a rate of ~ 9 μmol O$_2$ kg$^{-1}$ d$^{-1}$ from July 20-22, when the winds remained strong, and the OCR decreased to ~ 5.5 μmol O$_2$ kg$^{-1}$ d$^{-1}$ from Leg 2 to Leg 3 (from July 22-29) (Fig. S3). The OCR declined likely as inhibited by the reduced supply of labile organic matter (Yuan et al., 2010) — the strong blooms shifted offshore with the river plume and the Chl *a* concentrations over the hypoxic zone also decreased from Leg 2 to Leg 3 (Fig. 3k, l). If we considered the shoreward intrusion of shelf benthic waters following up the hypoxia development instead of contributing to lowering the initial DO level, their intrusion onto the inner shelf supplied oxygen to the oxygen-depleted waters, such as the cold, saline subsurface waters occupied the bottom layer covering the hypoxic zone on the inner shelf during Leg 1 (Fig. 4a, g). As density stratification limited the vertical diffusivity of oxygen from the upper layer, bottom hypoxia was therefore progressively formed and maintained until the decreasing OCR almost achieved equilibrium with the oxygen supply from the lateral advection.

### 4.2.3 Hypoxia formation timescale

The above estimated OCR and the contribution of shoreward-intruded shelf benthic waters made it possible to roughly estimate the hypoxia formation timescale: i.e., the time that the DO level in a known volume of water takes to decrease below the hypoxia threshold from an assumed initial DO concentration (Fennel and Testa 2019). In this study, for the reinstatement of hypoxia after the typhoon, the initial DO level in subsurface waters could be taken as $\sim 180$ µmol kg$^{-1}$, which varied within $\pm 5$ µmol kg$^{-1}$ throughout the time-series observations (Fig. 5c). Considering the OCR for the hypoxic zone was at most $\sim 20$ µmol O$_2$ kg$^{-1}$ d$^{-1}$, and the negligible contribution of lateral advection to oxygen loss off the Modaomen sub-estuary (Fig. 4), it took nearly 6 days for the drawdown of DO to reach concentrations of $\sim 63$ µmol kg$^{-1}$ within a limited area. Scaling to a larger area, it would instead take 8-12 days if we choose the average OCR of the oxygen-deficient zone, $\sim 15\pm 5$ µmol O$_2$ kg$^{-1}$ d$^{-1}$ (Fig. 8c). The closeness of these estimates with the water residence time ($\sim 15$ days, Li et al., (2020)) could partly explain the occurrence of periodic hypoxia to the west off the PRE (Su et al., 2017; Zhao et al., 2020).

For a more common scenario of hypoxia formation starting from late spring, we assumed that the shoreward-intruded oxygen-deficient offshore subsurface waters almost occupied the bottom layer on the inner shelf instead of the well-oxygenated offshore surface waters, as during Leg 1 (Fig. 4a, d). Based on above estimates that the shoreward-intruded subsurface waters totally reduced the initial DO level by $8.6\pm 1.7$ % of the oxygen decline for hypoxia formation, the initial DO level was lowered by $\sim 11$ µmol kg$^{-1}$ to be $\sim 183$ µmol kg$^{-1}$ before biochemical oxygen consumption. The hypoxia hotspot will then first occur $\sim 6$ days after its initiation on the inner NSCS shelf off the PRE and extend to a larger hypoxic zone within the water residence time of $\sim 15$ days (Lu et al., 2018; Li et al., 2020). This result is larger than that estimated by Fennel and Testa (2019) — 4 days — using the modelled OCR of $\sim 34$ µmol O$_2$ kg$^{-1}$ d$^{-1}$ in the water column with a sediment oxygen demand of $\sim 2.1$ g m$^{-2}$ d$^{-1}$, which were only applicable to the hypoxia formation in the Lingdingyang sub-estuary with shallower waters ($\sim 5$ m) (Zhang and Li 2010). This result is still at the lower end of the hypoxia formation timescale in large river-dominated shelves globally (e.g., East China Sea off the Changjiang estuary, Northern Gulf of Mexico and Northwestern Black Sea), which varies from 8 to 89 days for hypoxia to develop once initiated (Fennel and Testa 2019). This short hypoxia formation timescale likely owes to a high OCR in relatively warm subsurface waters fuelled by abundant labile organic matters (Su et al., 2017; Zhao et al., 2020).

### 4.3 Imprint of tropical cyclones on the evolution of coastal hypoxia

Tropical cyclones dramatically alter the physical stability of the water column and attenuate or even disrupt hypoxic conditions at low and mid-latitudes, such as off the Changjiang Estuary (Ni et al., 2016; Zhang et al., 2020) and the PRE (Su et al., 2017; Huang et al., 2019), in the Chesapeake Bay (Testa and Kemp 2014; Testa et al., 2017) and the northern Gulf of Mexico (Wang and Justić 2009; Feng et al., 2012). These intense, episodic storms thus strongly impact the duration and intensity of oxygen depletion in coastal bottom waters (Rabalais et al., 2009; Wang et al., 2017; Zhang et al., 2020), driving the seasonal hypoxia

to be intermittent. As shown in Fig. 1, at least four named tropical cyclones impacted the study area from May to August in 2018, most of which shifted the wind direction from prevailing southwesterly to easterly or southeasterly and increased the wind speed up to over 9 m s$^{-1}$, being able to destroy water column stratification and interrupt hypoxia formation (Geng et al., 2019). This was not an exception, as annually there were ~ 6 tropical cyclones travelling across the NSCS from May to September, when seasonal hypoxia develops (Qian et al., 2018; Wang et al., 2018), during the period from 1975-2019 (Fig.

9a). About five of the six annual tropical cyclones, on average, had the potential to overwhelmingly destroy the stability of the water column and replenish the bottom waters with oxygen. Indeed, when tropical cyclones impacted the NSCS over the last four decades, the local maximum wind speeds typically reached over 9 m s$^{-1}$, and often were larger than 15 m s$^{-1}$ (Fig. 9b); the local wind direction was inclined to be from the east, accounting for 56.2% in frequency, followed by the north (19.8%) and the west (16.0%) (Fig. 9d). The long-lasting easterly winds (Fig. 9e) were more likely to confine the river plume to the coast

(Xu et al., 2019) and even force the riverine freshwater to subduct down to the depth (Fig. S2), strengthening the reaeration of the oxygen-poor bottom waters (Wang et al., 2017; Wang et al., 2018; Huang et al., 2019). We also found that the time interval between two successive tropical cyclones was mostly less than 15 days, especially from July to September (Fig. 9c), close to the timescale for hypoxia formation. This might partly explain less hypoxia was formed and even observed before the year of 2000 (Yin et al., 2004; Rabouille et al., 2008). Therefore, frequent disturbance by tropical cyclones is one of the vital controls

on the intermittent hypoxia in low-latitude river-dominated ocean margins.

Although strong winds subdue hypoxia, tropical cyclones potentially promote intensive oxygen depletion after the transient dissipation of hypoxia (Rabalais et al., 2009). Heavy precipitation delivered by storms usually increases riverine freshwater loading to the coastal ocean (Zhou et al., 2012), resulting in intensified stratification when winds weaken (Wilson et al., 2008;

Su et al., 2017). Indeed, heavy precipitation happened in the Pearl River Delta region after the landing of typhoon SONTIHN (Guangdong Meteorological Service, http://gd.cma.gov.cn/qxfw/qhgk/201812/t20181217_92844.html), leading to a wide spreading of the river plume with lower salinity (< 15) than that during Leg 1 and the intensification of stratification (Fig. 6), even though the river discharge of the PRE increased insignificantly (Li et al., 2021). Enhanced vertical mixing and/or freshwater discharge supplied large amounts of nutrients to the surface layer to fuel phytoplankton blooms following large

storms (Zhao et al., 2009; Ni et al., 2016; Wang et al., 2017), as shown in Fig. 3 that strong blooms occurred in the surface plume along the coast with much higher Chl *a* concentrations during Leg 2 than that during Leg 1. The fresh autochthonous organic matter, together with the resuspended sedimentary organic carbon, provides sufficient substrates for microbial respiration in a re-stratified water column, leading to renewed or even exacerbated bottom water oxygen depletion (Zhou et al., 2012; Song et al., 2020). Along the east coast of North America, lowered DO concentrations were observed after storms

(Paerl et al., 2000; Tomasko et al., 2006). Hypoxia was re-established across a larger area when Hurricane Katrina crossed the southeast Louisiana coast (Rabalais et al., 2009). Off the PRE, we also found that hypoxia re-occurred in the wake of a more extensive freshwater plume and enhanced eutrophication after the passage of typhoon SONTIHN (Fig. 4). Whether it can

develop into more severe hypoxia compared to that found initially during Leg 1 depends on the net OCR and water column stability, up until the passage of the next storm, Typhoon BEBINCA (Fig. 1d).


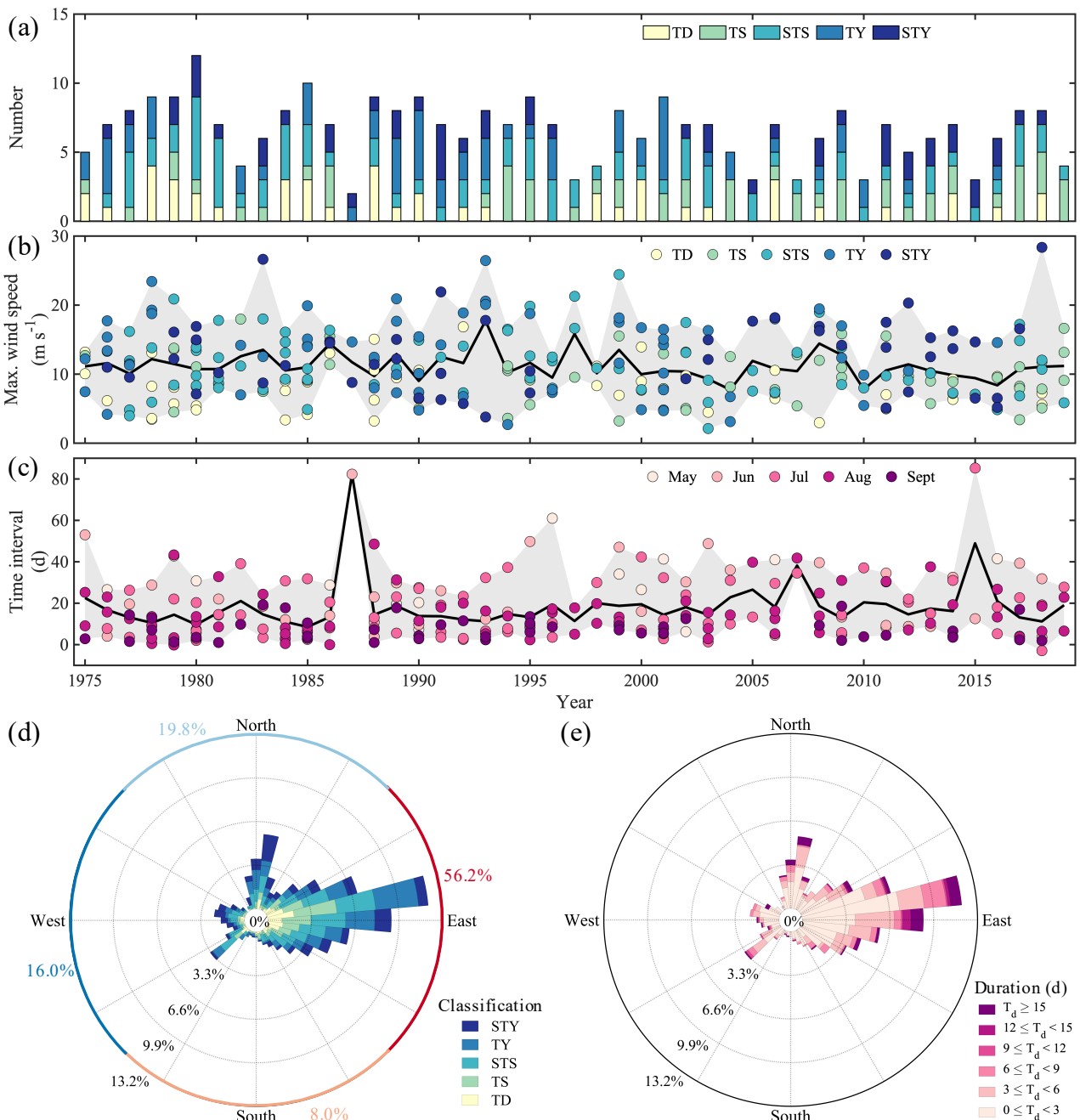

**Figure 9:** Statistics of tropical cyclones passing the northern South China Sea (NSCS) from May to September over 1975-2019. (a) Numbers of tropical cyclones. TD, TS, STS, TY and STY represent tropical depressions (the maximum wind speed near the centre is between 10.8-

17.1 m s$^{-1}$ over its lifetime), tropical storms (17.2-24.4 m s$^{-1}$), strong tropical storms (24.5-32.6 m s$^{-1}$), typhoons (32.7-41.4 m s$^{-1}$) and strong typhoons (41.5-50.9 m s$^{-1}$), respectively. (b) The maximum wind speed of each tropical cyclone. The black line and grey shadow denote the annual average and range of the maximum wind speeds. (c) The time interval between two successive tropical cyclones. The black line and grey shadow denote the annual average and range of the time intervals. (d) The wind rose of the intensity of tropical cyclones. (e) The wind rose of the duration of tropical cyclones. The wind speed in (b) and wind direction in (d, e) were recorded at the Waglan Island station (Figure 1b).

In a changing climate, tropical cyclone activity is expected to shift towards stronger storms but with a decreasing trend in frequency (Knutson et al., 2010). In the NSCS, the annual mean number of tropical cyclones in each decade decreased from 1950-2019, but the number of strong typhoons — the maximum wind speed near the center is between 41.5-50.9 m s$^{-1}$ over its lifetime — increased in the last decade (Table 2). The time interval between two successive tropical cyclones therefore increased by 2-3 days in the last two decades than before the year of 2000 (Fig. 9c). The less-frequent disturbance in the stability of the water column by tropical cyclones and the elongated time interval between two successive tropical cyclones likely favour more persistent hypoxia. However, intensified tropical cyclones would destroy the hypoxia more completely. Despite stronger blooms after tropical cyclones as during Leg 2 (Fig. 3k), the winds shifting back to prevailing southwest monsoon might drive the blooms offshore (Fig. 3l), reducing the downward transport of fresh, labile organic matters to fuel enduring intensive oxygen consumption in subsurface waters. In this sense, tropical cyclones may have the potential to relieve the exacerbation of coastal hypoxia in a warmer ocean.

**Table 2:** Summary of average frequency of tropical cyclones in each decade from 1950-2019. TD, TS, STS, TY and STY represent tropical depressions (the maximum wind speed near the centre is between 10.8-17.1 m s$^{-1}$ over its lifetime), tropical storms (17.2-24.4 m s$^{-1}$), strong tropical storms (24.5-32.6 m s$^{-1}$), typhoons (32.7-41.4 m s$^{-1}$) and strong typhoons (41.5-50.9 m s$^{-1}$), respectively.

| Years | TD | TS | STS | TY | STY | SUM |
|---|---|---|---|---|---|---|
| 1950-1959 | 3.5 | 1.1 | 1.2 | 1.1 | 1.5 | 8.4 |
| 1960-1969 | 1.7 | 0.6 | 1.5 | 2.1 | 2.7 | 8.6 |
| 1970-1979 | 1.8 | 0.7 | 2.1 | 2.1 | 1.2 | 7.9 |
| 1980-1989 | 1.5 | 0.7 | 2.5 | 1.3 | 1.3 | 7.3 |
| 1990-1999 | 0.7 | 1.2 | 1.8 | 2 | 1.1 | 6.8 |
| 2000-2009 | 0.9 | 1.4 | 1.5 | 1.5 | 0.8 | 6.1 |
| 2010-2019 | 0.6 | 1.8 | 1.4 | 0.4 | 1.5 | 5.7 |

**5 Conclusions**

We have demonstrated the evolution of intermittent hypoxia in summertime as disturbed by the typhoon passage on the inner NSCS shelf off the PRE and examined the controls on maintenance, destruction and reinstatement of hypoxia in this dynamic river-dominated marginal system. Eutrophication-induced hypoxia off the PRE was exacerbated with an enlarged area of ~ 660 km$^2$ and the lowest ever recorded regional DO concentration of 3.5 μmol kg$^{-1}$ (~ 0.1 mg L$^{-1}$). Freshwater inputs suppressed

turbulent mixing induced by wind stress and/or tidal forcing and stabilized the water column, restricting the ventilation of the subsurface water and facilitating the formation and maintenance of hypoxia. We estimated for the first time the upwelling-induced oxygen decline and *in situ* OCR over the destruction and reinstatement of hypoxia, which took place on a time scale of 6-12 d. This hypoxia timescale is comparable with water residence time and the disturbance of hypoxia from frequent tropical cyclones or high-wind events throughout the summer season, which could largely explain the intermittent nature of hypoxia off the PRE. Despite the less-frequent disturbance from tropical cyclones and the elongated time interval between successive tropical cyclones, the elevated intensity of tropical cyclones and possible offshore-advected blooms after tropical cyclones may have the potential to relieve the exacerbation of coastal hypoxia in a warmer ocean.

**Data Availability**. Data for temperature, salinity, DO and Chl *a* are currently for review and will be available at National Earth System Science Data Sharing Infrastructure, National Science & Technology Infrastructure of China (http://www.geodata.cn) with DOI. The wind speeds and directions at the Waglan Island from May to August, 2018 were obtained from the Hong Kong Observatory (http://www.hko.gov.hk/tc/cis/climat.htm). The tidal heights at the Dawanshan gauge station near the Station F303 from May to August, 2018 were downloaded from the website (http://www.chinaports.com/tidal/). Information from the tropical cyclone database (1949-2019) was obtained from the China Meteorological Administration (http://tcdata.typhoon.org.cn/).

**Supplement**. Additional figures referenced in text: **Figure S1**. Air temperature at the Hong Kong Observatory in July 2018. **Figure S2**. Distributions of temperature, salinity, DO and Chl *a* concentrations at the middle layer off the PRE during Leg 1 prior to Typhoon, and during Legs 2 and 3 post-typhoon. **Figure S3**. Profiles of temperature, salinity and DO at station F303 at the end of time-series observations and during Leg 2 and Leg 3.

**Competing interests**. The authors declare that they have no conflict of interest.

**Author contribution**. YZ and MD are major contributors to the study's conception, data analysis and drafting the paper. KU contributed the sample collection and measurements of DO data. ZL contributed substantially to sample collections and data analysis of physical forcing. HL provided Chl *a* data, and JG provided CTD data. YL, JL, and FM also contributed significantly to cruise design, sample collections and/or data acquisition.

**Acknowledgements**. This research was funded by the Hong Kong Research Grants Council under the Theme-based Research Scheme (TRS) through project no. T21-602/16R and the Ministry of Science and Technology of China under the National Key Scientific Research Project through grant 2015CB954000. Y. Zhao was also supported by the China Scholarship Council through a joint PhD program scholarship (201906310060). We thank Xue Song, Zhiqiang Liu, Dou Li, Zhongya Cai, Zhirong Xu and Xuechao Wang for their assistance in sample collections; Liguo Guo, Shilei Jin and Weizhen Jiang for assisting with

sample measurements; Jianzhong Su for discussions and suggestions; and Richard Smith of Global Aquatic Research for assistance in English. We express our gratitude to two anonymous reviewers for their insightful comments and suggestions that help substantially improve clarity of the paper. The captain and the crew of R/V Haike 68 are acknowledged for their cooperation during the cruise.

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
