# Peer review of "Destruction and reinstatement of coastal hypoxia in the South China Sea off the Pearl River Estuary"

_Biogeosciences, 2020_

## Referee Comment (RC1) · Anonymous Referee #1 · 4 Nov 2020

General comments:

In their manuscript Zhao et al investigate the effects of a typhoon on subsurface water reoxygenation following hypoxic condition off the Pearl River Estuary and the subsequent time scale of hypoxia formation. The author calculate the oxygen consumption rate in bottom waters using a three end-member mixing model. The subject of the paper is not novel but is an interesting approach and relevant for the region and the journal. The use of the model is interesting although the assumption about the "quasi-static bottom waters" is questionable and do not seem to fit with the observations (e.g. Figure 3, see specific comments below). The observations and analysis are fine but

too limited to fully support the conclusions. The authors rely heavily on the literature, whereas they could provide analysis or observations to support their interpretations. For example it would be interesting to see a sequence of oxygen vertical profiles, at multiple locations, when they look into the time-scale of hypoxia development after the typhoon. Volume might be an important factor at this time, although it is not mentioned because, as I understand, OCR is assumed to be uniform in subsurface waters. Does that imply that sediment oxygen demand is negligible? DIC observations are not used presently, although their analysis could strengthen the conclusions by relating OCR with respiration. Finally, part of the Discussion is a review of the literature (e.g. section 4.3) that is interesting but somewhat disconnected from the results presented. The discussion about hypoxia under future climate is highly hypothetical. Overall, the manuscript provides interesting results but could be significantly improved with more in-depth analyses of the observations. The quality of some figures could also be improved. Detailed comments are listed below.

Specific comments

L89: What are DIC and TA used for? How come you didn't use your DIC data to validate your estimate of OCR and to support your conclusions?

L101: did you collect chlorophyll samples? at what time of the day did you collect your chlorophyll profiles? did you notice non photochemical quenching near the surface and if so how did you correct the profiles?

Figure 1c: can you put shaded areas at the time of the cruises to be clear about the conditions during the sampling? Figure 1d: can you provide units and y-axis tick values? The length of the vectors don't seem to match the wind speed in the panel above. Also most vectors are oriented north-south and easterly winds vectors (associated with high wind speed) are small. Can you verify that the wind vectors are plotted properly?

L132: I have a hard time believing this assumption. Can you provide support to your claim? In Figure 3 bottom water conditions vary rapidly

Figure 3: This indicates the intrusion of warm/salty warters that participate to the re-stratification of the water column. What is the O2 and OM content of these waters? is OCR driven by sediment o2 consumption or just water column respiration? this figure indicates dynamic surface and bottom layers, not what it is described in the Results section (quiescent bottom layer). Do you know why O2 increases near the bottom? is there an advective source of O2 that is not taken into account?

L199: You should calculate stratification rather than citing others

L200: it didn't shift westward but was advected offshore indeed

L201: you are mixing discussion here

L241: but salinity (i.e. plume waters) will also control the wind intensity that is required to mix the water column

L246: Figure 2j,n shows stratification during leg 2, i.e. surface plume water, so the vertically homogeneous water column occurs before leg 2 (July 14-19)

L248: Figure 2l indicates a strong post-storm bloom that is not mentioned

L253: This should be true along the coast where the plume is trapped but not offshore

L254: This is discussed in a recent paper of the Changjiang estuary, you could have a look, I assume similar mechanisms occur in the PRE. https://doi.org/10.5194/bg-2019-341

L259: This is an interesting discussion but not supported by your observations so it feels a bit off topic

L291: what does that mean? that salinity was >xx in your samples?

L308: it is not a lower limit but an average estimate

L309: I am surprised that there was no advective sources/sinks of O2 given the observations in Figure 3
L313: You have to be clear how you use the terminology OCR. An increase in the OCR value (more positive) indicate a reoxygenetion.

L314: where is this shown?

L315: "decreased": so you mean less negative?

L317: you mean horizontal diffusion?

L316-320: your analysis is not well supported, can you discuss your results and be more quantitative? If not, please do not extrapolate

L324: Figure 3 indicates the intrusion of warm and salty subsurface waters at station F303 after the typhoon, can you comment on that and how it fits with your analysis?

L336: end of sentence: (Figure 5)

L338: those are really rough estimates. It is impossible to see what are the bottom O2 values in Figure 2 so it is difficult to judge your reasoning. Is the OCR=-15 value an average over the sampling area? bottom O2 values seem rather low over the entire area sampled during Leg 3

L341: Figure 3 shows that this is more variable than assumed here

L345: how did you come up to the value 183 from your assumption? do you assume mixing rather than water replacement during intrusion?

L346: This does not match your calculation above with the average OCR, why are you assuming the maximum OCR during spring?

L347: you should compare your values with similar systems, i.e. river-dominated estuaries (8-89 days). Also you could discuss your estimates in comparison of the PRE values provided in the reference.

L368-403: The last 2 paragraphs are not related to your results

L368: You did not mention/discussed the phytoplankton bloom in leg 2

[Figure]

L379: why?

Figure 6: I am not sure that annual averages are very pertinent, an average value per event might be more useful. For wind direction this could be presented as a pie chart. For wind speed, the time series is not very informative, may be think about an other way of presenting the results. This is somewhat included in Figure 6a but it would be interesting to know what is the maximum time between wind events for each year (or have some statistics based on your estimate of OCR). Also remember that hypoxia did not occur for most of the period shown here. panel c: please see comment regarding wind vectors in Figure 1, make sure those are right

L398: it depends on the direction, offshore intrusions would presumably bring lower O2 waters

L409: "lowest ever recorded"

L417: This is speculation, higher discharge may lead to lower nutrient concentrations in river waters, more export to deep areas where hypoxia does not occur

Minor comments/edits:

L80: can you provide the number of stations in parenthesis for each leg, it is difficult to estimate it from Figure 1b

L81: suggestion: "on the way back to port"

L155-164: You are mixing results and discussion

Figure 2: there is no point showing the river labels, they are way too small. Also the contour labels cannot be seen and the color bars are way too small. I suggest you move the colorbar to the top of each column and make it thicker with larger fonts An alternative suggestion is to split the figure into surface and bottom figures and flip the rows to columns to make larger panels

Figure 4: The labels and lines are very small

L312: not clear, the sentence should be rephrased
* * *

---

## Referee Comment (RC2) · Anonymous Referee #2 · 25 Nov 2020

Review of Zhou et al. Destruction and reinstatement of coastal hypoxia in the South China Sea off the Pearl River Estuary. I found this manuscript to be a useful contribution to our understanding of the spatial and temporal nature of oxygen depletion as q large coastal system responds to large events. The narrative is relatively easy to follow and the results are clearly communicated with figures. I think the analysis could benefit from a small amount of additional computations, but I also think that the results and discussion section needs to be reorganized. There is substantial mixing of results and discussions between the two sections, and I think it would be best and easiest to simply combine the two sections into one "Results and Discussion" section that is re-organized into a clear narrative. Below are some specific and more general comments

[Figure]

for the authors to consider:

(1) Line 49: "typhoons" should be plural

(2) Figure 1: It is a little difficult to discern the station locations of the different legs, given the overlapping in the circles. One suggestion could be to use different symbols to present (1) stations visited on all legs, (2) stations visited on legs 1+2, (3) stations visited on legs 2+3, and (4) stations visited on 1+3.

(3) Although Figure 3 nicely illustrates how stratification returned after the cyclone, it does not capture any patterns over space and it does not capture the entire coverage of the study in time. Figure 2 provides a nice, qualitative picture of the changes in water properties over time and space, but I think it might be helpful to also generate maps of the stratification changes, perhaps by plotting max N2 over space or the difference in temperature and salinity (or density) between surface and bottom waters. This would have the benefit of showing if stratification was weaker after it was reinstated than before the typhoon, where stratification was strongest, and how it related in space to hypoxia.

(4) Figure 4 – I think it would be more interesting to also show vertical oxygen distributions on figure 4, to show where hypoxia exists relative to the vertical structure and stratification.

(5) You report on the decline in oxygen concentration in the water column after the typhoon passing as a metric of oxygen consumption rate. It would make the paper more compelling, and help the discussion, to compare these rates of oxygen depletion to similar rates published in other systems (e.g., Testa and Kemp 2014, others?)

(6) Line 315-319: Can you estimate the oxygen diffusivity rate from your data, based on any published estimates of diffusivity for the region, or estimated from your density profiles? This would allow you to be more quantitative in your comparison of OCR and diffusivity as eventually balancing. I think you could also speculate, perhaps with data,

why OCR could have possibly declined, either as the post-bloom organic material was exhausted or due to oxygen limitation of respiratory uptake?

(7) Paragraph on Line 389: This paragraph reads more like an essay on the factors driving hypoxia and vulnerable to climate change, and does not really discuss the specific details of this study. I suggest deleting it, perhaps keeping the cyclone points for the prior paragraph on cyclone effects.

(8) I think you should combine the Results and Discussion Sections into one, well-organized narrative. As it stands, there are multiple places where results are reported in the discussion, or there are even methods in the discussion. This would allow you to more clearly and sequentially tell the story of your study. Below are some specific examples to guide this effort: (a) Line 225-239 is largely results and even methods, but is included in the discussion without substantial discussion of the results in the context of the study. (b) Line 285-290. Here, you are describing the method you already described. Move to methods and remove redundancy. (c) Paragraphs beginning on lines 332 and 343 can be combined

---

## Author Comment (AC1) · 29 Dec 2020

The comment was uploaded in the form of a supplement:
https://bg.copernicus.org/preprints/bg-2020-346/bg-2020-346-AC1-supplement.pdf

---

## Author Comment (AC2) · 29 Dec 2020

The comment was uploaded in the form of a supplement:
https://bg.copernicus.org/preprints/bg-2020-346/bg-2020-346-AC2-supplement.pdf

---

## Author Response (AR1)

*Re: Destruction and reinstatement of coastal hypoxia in the South China Sea off the Pearl River Estuary" by Yangyang Zhao et al.*

January 31, 2021

Dear Editor,

Thank you for your time in handling our paper. Enclosed please find our revised MS entitled "Destruction and reinstatement of coastal hypoxia in the South China Sea off the Pearl River Estuary" by Yangyang Zhao et al.

During the revisions, we have fully considered the comments and suggestions from the reviewers. Briefly, we elaborated the dynamics of the time-series observations. We followed the reviewer's suggestion and calculated the biochemical-induced DIC changes from Leg 2 to Leg 3 and their relationship to the biochemical-induced oxygen changes and the vertical diffusion for oxygen based on the published diffusivity from Cui et al. (2019), to support our estimates of oxygen consumption rates (OCR). We also compared our estimates of the OCR and hypoxia formation timescale with previous studies in this study area and other large river-dominated shelf systems. Taken together, this study provides a robust approach to estimate the *in situ* OCR for hypoxia formation at large-scales.

Following suggestions, we have restructured our MS by combining results and discussion into a clear narrative. Moreover, we have expanded our discussion and included the tidal effects on hypoxia, the impacts of tropical cyclones-induced processes on hypoxia restoration, and the response of the hypoxia's evolution to the changes in frequency and intensity of tropical cyclones. Additionally, we have improved the quality of the figures for better illustrations. More detailed revisions are explained in the enclosure.

Finally, we would like to take this opportunity to thank the reviewers for their constructive comments and suggestions, which significantly improved the quality of the paper. We sincerely hope that our revision will meet the standards of *Biogeosciences*.

Sincerely,

Minhan Dai

Corresponding author

State Key Laboratory of Marine Environmental Science

Xiamen University

Xiamen 361005, China

Phone: 86-592-218-2132

Fax: 86-592-218-0655

E-mail: mdai@xmu.edu.cn

**Anonymous Referee #1**

**General comments:**

*In their manuscript Zhao et al investigate the effects of a typhoon on subsurface water reoxygenation following hypoxic condition off the Pearl River Estuary and the subsequent time scale of hypoxia formation. The author calculate the oxygen consumption rate in bottom waters using a three end-member mixing model. The subject of the paper is not novel but is an interesting approach and relevant for the region and the journal. The use of the model is interesting although the assumption about the "quasi-static bottom waters" is questionable and do not seem to fit with the observations (e.g. Figure 3, see specific comments below). The observations and analysis are fine but too limited to fully support the conclusions. The authors rely heavily on the literature, whereas they could provide analysis or observations to support their interpretations. For example it would be interesting to see a sequence of oxygen vertical profiles, at multiple locations, when they look into the time-scale of hypoxia development after the typhoon. Volume might be an important factor at this time, although it is not mentioned because, as I understand, OCR is assumed to be uniform in subsurface waters. Does that imply that sediment oxygen demand is negligible? DIC observations are not used presently, although their analysis could strengthen the conclusions by relating OCR with respiration. Finally, part of the Discussion is a review of the literature (e.g. section 4.3) that is interesting but somewhat disconnected from the results presented. The discussion about hypoxia under future climate is highly hypothetical. Overall, the manuscript provides interesting results but could be significantly improved with more in-depth analyses of the observations. The quality of some figures could also be improved. Detailed comments are listed below.*

**[Response]**: We appreciate the critical and constructive comments from the reviewer. The reviewer is correct that bottom waters may be dynamic upon disturbance by typhoons as reflected notably in vertical mixing. Here our "quasi-static bottom waters" assumption was referring to the limited exchanges between the oxygen-depleted bottom water mass under study with the surrounding environment. More specifically, we contend that this assumption was reasonable for calculating the oxygen consumption rate (OCR) from Leg 2 to Leg 3 because hypoxic zones typically developed in strong convergent zones (Lu et al., 2018; Li et al., 2020) with thus longer residence time that facilitates oxygen consumption. Indeed, the bottom water residence time in the sampling area was ~ 15 days (Li et al., 2020), significantly longer than the time lag from Leg 2 to Leg 3 (~ 6 days). We thus contend that the disturbance by typhoon mainly led to vertical mixing instead of horizontal water mass exchanges. The upward-intruded bottom waters were indeed visible in the time-series observations, and such vertical mixing

was gradually suppressed by the surface plume-induced stratification (Fig. R1). The water column below the pycnocline became almost vertically well-mixed towards the end of the time-series observations before Leg 2. Based on the above notions and the reviewer's comments, we have revised our assumption as "the bottom water masses where biochemical oxygen-consumption prevailed were constrained by strong convergence and their outflow from the sampling area is insignificant on the time scale of the water residence time" in our revised manuscript (Page 6, Lines 142-144).

The reviewer is also correct that the OCR was assumed to be uniform in the subsurface waters, because (1) we only collected samples at three depth layers during Leg 2 and Leg 3, with usually only one depth layer below the pycnocline; (2) if the middle layer was below the pycnocline, the concentration of dissolved oxygen (DO) almost equaled to that at the bottom layer (e.g., Station F304; Fig. R3k). Sediment oxygen demand might be significant near the seabed or in its overlying water column (Kemp et al., 1992; Zhang and Li 2010). Here in our sampling area, oxygen losses by sediment oxygen demand (i.e., benthic respiration) were found much smaller than the bacterial respiration in the water column based on both incubation experiments and oxygen budget analysis (Cui et al., 2019). We thus assumed that the sediment oxygen demand was negligible and the microbial respiration in the water column dominated the estimated OCR (Page 19, Lines 434-442 of our revised MS).

We agree with the reviewer that we should fully use our data to support our conclusions. Dissolved inorganic carbon (DIC) and total alkalinity (TA) were used to validate our three-endmember mixing model. We have further calculated the biochemical-induced DIC changes from Leg 2 to Leg 3 and showed their relationship to the biochemical-induced DO changes, with a slope of -0.92±0.17 (Fig. R2), implying aerobic respiration dominated the OCR (Page 18, Lines 409-411 of our revised MS). We must point out that we did not directly estimate the DIC production rate. This is because uncertainties in DIC predications from the conservative mixing among three water masses reached $\sim$ 30 µmol kg$^{-1}$, comparable to changes in DIC between two legs for nearly half of the sampling stations during Leg 3. We have however further added the oxygen profiles at multiple stations to further illustrate their temporal variations (Fig. R3): the DO concentration below the pycnocline significantly decreased from Leg 2 to Leg 3 and was almost homogeneous vertically (Page 14, Lines 302-305 of our revised MS).

Following comments, we have also reorganized our paper by combining results and discussion into a clear narrative, with an outline as below:

3 Evolution of intermittent hypoxia off the PRE

        3.1 Extensive hypoxia before the typhoon

        3.2 Destruction of hypoxia by the typhoon

        3.3 Reinstatement of hypoxia after the typhoon

4 Maintenance, destruction and reinstatement of coastal hypoxia

        4.1 Water column stability

        4.2 Oxygen sinks and hypoxia formation timescale

            4.2.1 Mixing-induced oxygen sinks

            4.2.2 Biochemical-induced oxygen sinks

            4.2.3 Hypoxia formation timescale

        4.3 Imprint of tropical cyclones on the evolution of coastal hypoxia

In particular, we have further modified the discussion, including the tidal effects on hypoxia, the impacts of tropical cyclones-induced processes on hypoxia restoration, and the response of the hypoxia's evolution to the changes in frequency and intensity of tropical cyclones. We have calculated the potential maximum hypoxic area as spring-to-neap tidal oscillations lead to variations in the DO concentration with a maximum neighboring oxygen range of 0.5 mg $L^{-1}$ (Cui et al., 2019) (Page 16, Lines 345-360 of our revised MS). We have correlated post-storm precipitation and river discharge to our observations showing post-storm stronger blooms (Pages 21-22, Lines 512-529 of our revised MS). We have also added statistics on annual mean number and wind direction of tropical cyclones and the time interval between two successive tropical clones (Fig. R4 and Table R1) (Pages 23, Lines 541-551 of our revised MS).

Finally, we have improved the quality of the figures to better illustrate spatial distributions of temperature, salinity, DO and chlorophyll *a* (Chl *a*) concentrations (Pages 10, Figure 3 and Page 11, Figure 4 of our revised MS). We will further address these concerns from the reviewer in our responses as of below.

[Figure]

**Figure R1 :** Time-series observations of (a) temperature (°C), (b) salinity, (c) DO (µmol kg⁻¹) and (d) Chl *a* concentrations (µg L⁻¹) at Station F303 (see Fig. 1b) from July 19-20, 2018 after the typhoon passage, showing the complete destruction and the subsequent rapid development of stratification.

[Figure]

**Figure R2:** Biochemical-induced changes in DIC ($\Delta DIC^{bc}$, µmol kg⁻¹) vs. DO ($\Delta DO^{bc}$, µmol kg⁻¹) in bottom waters with depths > 10 m from Leg 2 to Leg 3. The black line denotes the slope of $\Delta DIC^{bc}$ plotted against $\Delta DO^{bc}$ derived from the Model II regression.

[Figure]

**Figure R3:** Profiles of temperature (°C) (green dashed lines), salinity (pink solid lines), dissolved oxygen (DO, μmol kg$^{-1}$) (purple dots) and buoyancy frequency $N^2$ (s$^{-2}$) (bold black solid lines) at stations A8, A11, A14 and F304 (see Fig. 1b), with visits both pre-typhoon (Leg 1) and post-typhoon (Legs 2 and 3). The vertical distributions of $N^2$ have been smoothed by the Gaussian method.

[Figure]

**Figure R4:** Statistics of tropical cyclones passing the northern South China Sea (NSCS) from May to September over 1975-2019. (a) Numbers of tropical cyclones. TD, TS, STS, TY and STY represent tropical depressions (the maximum wind speed near the centre is between 10.8-17.1 m s$^{-1}$ over its lifetime), tropical storms (17.2-24.4 m s$^{-1}$), strong tropical storms (24.5-32.6 m s$^{-1}$), typhoons (32.7-41.4 m s$^{-1}$) and strong typhoons (41.5-50.9 m s$^{-1}$), respectively. (b) The maximum wind speed of each tropical cyclone. The black line and grey shadow denote the annual average and range of the maximum wind speeds. (c) The time interval between two successive tropical cyclones. The black line and grey shadow denote the annual average and range of the time intervals. (d) The wind rose of the intensity of tropical cyclones. (e) The wind rose of the duration of tropical cyclones. The wind speed in (b) and wind direction in (d, e) were recorded at the Waglan Island station.

**Table R1:** Summary of average frenquency of tropical cyclones in each decade from 1950-2019. TD, TS, STS, TY and STY represent tropical depressions (the maximum wind speed near the centre is between 10.8-17.1 m s$^{-1}$ over its lifetime), tropical storms (17.2-24.4 m s$^{-1}$), strong tropical storms (24.5-32.6 m s$^{-1}$), typhoons (32.7-41.4 m s$^{-1}$) and strong typhoons (41.5-50.9 m s$^{-1}$), respectively.

| Years | TD | TS | STS | TY | STY | SUM |
|---|---|---|---|---|---|---|
| 1950-1959 | 3.5 | 1.1 | 1.2 | 1.1 | 1.5 | 8.4 |
| 1960-1969 | 1.7 | 0.6 | 1.5 | 2.1 | 2.7 | 8.6 |
| 1970-1979 | 1.8 | 0.7 | 2.1 | 2.1 | 1.2 | 7.9 |
| 1980-1989 | 1.5 | 0.7 | 2.5 | 1.3 | 1.3 | 7.3 |
| 1990-1999 | 0.7 | 1.2 | 1.8 | 2 | 1.1 | 6.8 |
| 2000-2009 | 0.9 | 1.4 | 1.5 | 1.5 | 0.8 | 6.1 |
| 2010-2019 | 0.6 | 1.8 | 1.4 | 0.4 | 1.5 | 5.7 |

**Specific comments**

*L89: What are DIC and TA used for? How come you didn't use your DIC data to validate your estimate of OCR and to support your conclusions?*

**[Response]**: DIC and TA were used for validating the three-endmember mixing model. TA is a quasi-conservative parameter due to its small changes during biological processes. Comparing our predicted values with measured TA, we found they were consistent with a subtle difference of 8±8 μmol kg$^{-1}$, which is asscoiated with measurement errors, propagation of uncertainty through the mixing scheme and/or biological processes (Fig. R5b). DIC is produced with the oxygen depletion. We calculated the biochemical-induced DIC and DO changes for each leg and found that they had a good relationship with a slope of -0.93±0.07 (Fig. R5c), similar to that reported by Zhao et al. (2020) for the same study area (Page 7, Lines 174-177 of our revised MS).

Following suggestions, we calculated the biochemical-induced DIC changes ($\Delta DIC^{bc}$) from Leg 2 to Leg 3. We must point out that uncertainties in DIC predictions from the conservative mixing among three water masses reached ~ 30 μmol kg$^{-1}$, mainly due to a large variability in the DIC concentration of the brackish plume endmember (Table R2). These uncertainties were comparable to changes in DIC between two legs for nearly half of the sampling stations during Leg 3 (uncertainties in DO were only ~ 5 μmol kg$^{-1}$, much smaller than changes in the DO concentration from Leg 2 to Leg 3). We thus plotted the biochemical-induced DIC changes versus DO changes from Leg 2 to Leg 3, also showing a good relationship with a slope of -0.92±0.17 (Fig. R2), to indirectly validate our estimates of the OCR. This slope was consistent

throughout the sampling legs, implying aerobic respiration of organic matters indeed dominated the OCR (Page 18, Lines 409-411 of our revised MS).

[Figure]

**Figure R5**: (a) Potential temperature (°C) vs. salinity, (b) predicted TA (TA[pre], μmol kg[-1]) vs. measured TA (TA[meas], μmol kg[-1]), and (c) ΔDIC (μmol kg[-1]) vs. ΔDO (μmol kg[-1]) on the NSCS shelf off the PRE. The black-edged circles represent bottom water samples with depths > 10 m. The yellow, green and purple triangles in (a) represent the endmember values of Brackish Plume Water (PW), offshore surface water (SW) and upwelled subsurface water (SUB), respectively. The black line in (c) denotes the slope of ΔDIC plotted against ΔDO derived from the Model II regression.

**Table R2:** Summary of the end-member values adopted in the three-endmember mixing model

| Water mass | θ (°C) | Salinity | DIC (μmol kg[-1]) | DO (μmol kg[-1]) |
|---|---|---|---|---|
| Brackish plume water | 28.9±0.4[b] | 16.9 | 1776±29[b] | 217.3±1.4[c] |
| Offshore surface water[a] | 29.3±0.1 | 33.7±0.1 | 1922±5 | 194.4±0.3[c] |
| Upwelled subsurface water[a] | 22.5±0.1 | 34.5±0.0 | 2022±3 | 180.9 |

[a]Adopted from Zhao et al., (2020)
[b]Uncertainties were derived from mutiple samples collected at the entrance of the PRE
[c]Uncertainties were calculated by propagating errors associated with the estimation of oxygen solubility using Benson and Krause Jr (1984)

*L101: did you collect chlorophyll samples? at what time of the day did you collect your chlorophyll profiles? did you notice non photochemical quenching near the surface and if so how did you correct the profiles?*

**[Response]**: We collected samples for chlorophyll *a* (Chl *a*) concentrations along with the cruise track during both day and night. We noticed the non-photochemical quenching near the surface from the fluorescence sensor, which was however not used to derive Chl *a*. Instead, we obtained the Chl *a* concentrations from discrete water samples. These samples were filtered onto GF/F (Whatman, USA) and stored in foil bags in liquid nitrogen until they were measured on a Trilogy laboratory fluorometer (Welschmeyer 1994) after extracted with 90% acetone for 14 h at -20 °C (Page 3, Lines 88-91 and Page 4, Lines 99-101 of our revised MS).

*Figure 1c: can you put shaded areas at the time of the cruises to be clear about the conditions during the sampling? Figure 1d: can you provide units and y-axis tick values? The length of*

*the vectors don't seem to match the wind speed in the panel above. Also most vectors are oriented north-south and easterly winds vectors (associated with high wind speed) are small. Can you verify that the wind vectors are plotted properly?*

**[Response]**: We appreciate the suggestions. Accordingly, we have added shaded areas at the time of the cruises for each leg and re-plotted Figure 1d to match the arrow lengths with the wind speed (Fig. R6; Page 5, Figure 1 of our revised MS). During the cruise legs, easterly winds dominated with relatively larger east-west components than south-north components of wind velocities.

[Figure]

**Figure R6**: (a) Map of the study area on the shelf of the northern South China Sea (NSCS), showing the track of Typhoon SONTIHN (circles) across the NSCS during July 16-24, 2018. The color of the circles represents the magnitude of wind speed. Additionally, the smaller circles denote tropical depression (wind speeds ≤ 17.1 m s⁻¹) and the larger circles denote tropical storm (wind speeds within 17.2-32.6 m s⁻¹). The arrows denote the locations of the typhoon as marked with time and wind speed. The grey lines are the depth contours at 50 and 200 m. (b)

Sampling stations on the NSCS shelf off the Pearl River Estuary in summer 2018. The pink, green, purple and orange circles denote the stations surveyed in all three legs, only both Leg 1 and Leg 2, only Leg 1 and only Leg 2, respectively. Time-series observations were conducted at Station F303 as marked by the star, and vertically high-resolution samplings were conducted at stations marked with bold circles. (c) The wind speed and (d) wind direction at Waglan Island (triangle in (b)) from May to August, 2018. Bars at the bottom of (d) mark times when tropical cyclones impacted the NSCS. (e) The tidal height at the Dawanshan gauge station near Station F303 from May to August, 2018. The shaded area indicates the cruise periods for Leg 1 (grey), Leg 2 (pink) and Leg 3 (blue), respectively.

*L132: I have a hard time believing this assumption. Can you provide support to your claim? In Figure 3 bottom water conditions vary rapidly*

**[Response]**: We appreciate the reviewer's comment, which is indeed critical to this study. Please refer to our response to the general comment (Pages 1-2 of this response).

*Figure 3: This indicates the intrusion of warm/salty warters that participate to the re-stratification of the water column. What is the $O_2$ and OM content of these waters? is OCR driven by sediment $O_2$ consumption or just water column respiration? this figure indicates dynamic surface and bottom layers, not what it is described in the Results section (quiescent bottom layer). Do you know why $O_2$ increases near the bottom? is there an advective source of $O_2$ that is not taken into account?*

**[Response]**: We appreciate the critical comments. This figure indeed showed the intrusion of warm/salty waters, with the DO concentrations of ~ 180-184 µmol kg$^{-1}$ and the POC concentrations of 0.12-0.16 mg L$^{-1}$, which differed by < 4 µmol kg$^{-1}$ and < 0.04 mg L$^{-1}$, respectively, from the upper waters. The estimated OCR here was mainly driven by water column respiration detailed in our response above (Page 2 of this response).

We agree with the reviewer that the surface and bottom layers were dynamic over time, but with quite small variations (Fig. R1). As described in the Results section, DO in the bottom layer was homogeneous and temperature and salinity showed smaller cross-shore gradients than those during Leg 1, rather than quiescent. The slightly higher DO near the bottom layer likely resulted from previous strong vertical mixing upon disturbance by the typhoon, which mixed high-DO surface waters into the depth. As the upward-intruded bottom waters in our time-series observations were warm but salty, and the DO distributions were almost homogeneous at the bottom layer during Leg 2 (Fig. R7), these bottom waters unlikely sourced from nearshore warm, less-salty waters or offshore cooler, saline waters via lateral advection. We thus contend that we have taken into account all advective sources of oxygen from Leg 2 to Leg 3.

[Figure]

**Figure R7:** Distributions of temperature (°C), salinity, DO (µmol kg⁻¹) and Chl *a* concentrations (µg L⁻¹) at the bottom water layer off the PRE during Leg 1 pre-typhoon, and during Legs 2 and 3 post-typhoon. The white and magenta contours in (g) and (w) show the hypoxic (DO < 63 µmol kg⁻¹) and oxygen-deficit (DO < 94 µmol kg⁻¹) zones.

*L199: You should calculate stratification rather than citing others*

**[Response]**: We appreciate the suggestion. We calculated stratification using the buoyancy frequency (Page 4, Lines 102-109 of our revised MS), but we cannot partition the contributions from the vertical gradients in temperature or salinity. We cited the reference here to support that vertical temperature gradients could intensify stratification in addition to vertical salinity gradients. We therefore would like to keep this citation as we have reorganized the paper by combining results and discussion into a clear narrative with an outline listed above in our response to the general comment (Page 3 of this response).

*L200: it didn't shift westward but was advected offshore indeed*

**[Response]**: Accepted. By carefully comparing the surface salinity distributions during three legs, we have corrected the statement as "The freshwater bulge of lower salinity (< 15) advected offshore around the Modaomen sub-estuary" in our revised manuscript (Page 9, Lines 250-251).

*L201: you are mixing discussion here*

**[Response]**: Accepted. We have reorganized the paper by combining results and discussion to a clear narrative with an outline as listed above in our response to the general comment, and also revised the statement as "… likely driven by the interaction between the seaward buoyant current and northeastward shelf current (Pan et al., 2014; Li et al., 2020)" in our revised manuscript (Page 9, Lines 252-253).

*L241: but salinity (i.e. plume waters) will also control the wind intensity that is required to mix the water column*

**[Response]**: We agree with the reviewer that the wide-spreading plume also influences the wind-driven turbulent mixing. We discussed in sequence the effects of freshwater inputs (i.e., river plume), wind stress and direction, tidal fluctuations and spring-to-neap tidal oscillations on the water column stability. We therefore have corrected the statement as "Water column stability also largely depends on wind stress in coastal waters" and discussed the effects of river plume vs. wind stress on the water column stability in our revised manuscript (Pages 14-15, Lines 317-340).

*L246: Figure 2j,n shows stratification during leg 2, i.e. surface plume water, so the vertically homogeneous water column occurs before leg 2 (July 14-19)*

**[Response]**: Accepted. We have corrected the statement as "The strong winds facilitated mixing between high-temperature, low-salinity surface waters and cold, saline bottom waters, resulting in a vertically-homogeneous temperature and salinity, as observed in the first half of the time-series observations before Leg 2" in our revised manuscript (Page 14, Lines 321-323).

*L248: Figure 2l indicates a strong post-storm bloom that is not mentioned*

**[Response]**: We described the strong post-storm bloom in Section 3.2 — Destruction of hypoxia by typhoon — "Stronger blooms than that during Leg 1 were identified in the surface plume, widely spreading from the mouth of the Lingdingyang sub-estuary to near the Huangmaohai sub-estuary, potentially fueled by nutrients mixed upward from the deep in

addition to riverine inputs (Wang et al., 2017; Qiu et al., 2019). The maximum Chl *a* concentration was > 40 μg $L^{-1}$ off the Modaomen sub-estuary, accompanied by an extraordinarily high DO concentration of > 350 μmol $kg^{-1}$" (Page 9, Lines 228-232 of our revised MS).

*L253: This should be true along the coast where the plume is trapped but not offshore*
**[Response]**: We agree with the reviewer. Indeed, here we discussed the water column stability along the coast mostly within the 30-m isobaths where the plume was trapped.

*L254: This is discussed in a recent paper of the Changjiang estuary, you could have a look, I assume similar mechanisms occur in the PRE. https://doi.org/10.5194/bg-2019- 341*
**[Response]**: We appreciate the suggestion. Zhang et al., (2020) discussed the effect of wind direction on the plume spreading: the strong northward wind redistributes and advects the river plume towards the Yangtze Bank through Ekman transport, while the weakened northward or westward wind allows the location of the bottom hypoxia to migrate to the Submarine Canyon (Zhang et al., 2019). Similarly, off the PRE, upwelling-favorable winds (i.e., southwesterly) drive the river plume offshore and eastward (Gan et al., 2009), increasing the hypoxia area. However, downwelling-favorable easterly winds tend to constrain the river plume near the coast and drive surface waters to penetrate into the depth (Fig. R7), leading to an offshore shift of the hypoxic zone within a limited area beneath the surface plume. If the easterly winds last for a longer time than the hypoxia timescale, stronger blooms in the surface plume would enhance the bottom hypoxia with an abundant supply of fresh, labile organic matters. Based on the above notions, we have revised the statement in our revised manuscript (Pages 14-15, Lines 331-340).

*L259: This is an interesting discussion but not supported by your observations so it feels a bit off topic*
**[Response]**: Accepted. We have revised this discussion by calculating the potential maximum area of ~ 990 $km^2$ and ~ 1930 $km^2$ for the hypoxic and oxygen-deficient zones. The calculation is based on that spring-to-neap tidal oscillations lead to variations in the DO concentration with a maximum neighboring oxygen range of 0.5 mg $L^{-1}$ (Cui et al., 2019) and our cruise legs were all conducted during the transformation from a neap tide to a spring tide (Fig. R6e). The hypoxia and oxygen-deficient areas therefore might be underestimated by at most of 34-50% in our sampling area due to tidal fluctuations (Page 16, Lines 345-360 of our revised MS).

*L291: what does that mean? that salinity was >xx in your samples?*

**[Response]**: The statement "we selected samples with water depths > 10 m, approximately below the pycnocline and where the surface plume was rarely involved" means that samples used for calculating the OCR were below the pycnocline and less affected by the upper plume waters. The selected samples with water depths > 10 m have salinity of > 31. We have revised the statement as "In estimating the OCR, we excluded the above-pycnocline samples collected at depths < 10 m affected by the upper plume waters that are subject to strong air-sea exchanges and/or photosynthetic production of oxygen" in our revised manuscript (Page 7, Lines 171-172).

*L308: it is not a lower limit but an average estimate*

**[Response]**: The OCR was spatially averaged, but it was also a lower limit. This is because we might overestimate the actual time for the significant oxygen consumption since its initiation between Leg 2 and Leg 3 and underestimate the diffusion of oxygen from the surface layer (assumed negligible during our estimates of the OCR). Both factors likely lead to an underestimation of the OCR (Pages 18-19, Lines 429-434 of our revised MS).

*L309: I am surprised that there was no advective sources/sinks of $O_2$ given the observations in Figure 3*

**[Response]**: We admit that there were vertically advective sources/sinks of $O_2$ in the time-series observations (Fig. R1). However, the upward intrusion of slightly warm/salty waters was progressively suppressed by the surface plume towards the end of the time-series observations. We also observed the wide-spreading river plume at the surface layer during Leg 2 (Fig. R8), restoring the vertical stratification and restricting the oxygen supply from the surface layer. In addition, the slightly higher DO near the bottom layer likely resulted from previous strong vertical mixing upon disturbance by typhoon, which mixed high-DO surface waters into the depth. We found that the upward-intruded bottom waters in our time-series observations were warm but salty, and the DO distributions were almost homogeneous at the bottom layer during Leg 2 (Fig. R7). These bottom waters were thus unlikely sourced from nearshore warm, less-salty waters or offshore cooler, saline waters via lateral advection. We contend that we have taken into account all advective sources of oxygen from Leg 2 to Leg 3.

[Figure]

**Figure R8:** Distributions of temperature (°C), salinity, DO (μmol kg⁻¹) and Chl *a* concentrations (μg L⁻¹) at the surface water layer off the PRE during Leg 1 pre-typhoon, and during Legs 2 and 3 post-typhoon. The white and magenta contours in (g) and (w) show the hypoxic (DO < 63 μmol kg⁻¹) and oxygen-deficit (DO < 94 μmol kg⁻¹) zones. Figures were produced using Ocean Data View v. 5.3.0 (http://odv.awi.de, last access: 08 June 2020)

*L313: You have to be clear how you use the terminology OCR. An increase in the OCR value (more positive) indicate a reoxygenetion.*

**[Response]**: We appreciate the critical comment. Changes in the OCR of a negative value were confusing. We have revised the definition for the OCR as the biochemical-induced DO consumption with time. A higher OCR value indicates stronger oxygen consumption and a negative value indicates oxygen production due to biochemical processes (e.g., photosynthesis) (Page 4, Lines 113-115 of our revised MS). Accordingly, we have corrected all the statements

associated with changes in the OCR (Page 1, Line 18; Page 18, Lines 412-415; Page 19, Lines 452-454; Page 20, Lines 468-471 of our revised MS).

*L314: where is this shown?*

**[Response]**: The data were not shown in time series for explicit comparisons. We have plotted the figure to show the data (Fig. R9) and have added this figure to our revised supplementary material (Page 4, Figure S3 of our revised Supplement).

[Figure]

**Figure R9:** Profiles of temperature (°C), salinity and DO ($\mu$mol kg$^{-1}$) at station F303 at the end of the time-series observations and during Leg 2 and Leg 3.

*L315: "decreased": so you mean less negative?*

**[Response]**: The reviewer is correct that the OCR was less negative as it varied from ~ -9 $\mu$mol O$_2$ kg d$^{-1}$ (July 20-22) to ~ -5.5 $\mu$mol O$_2$ kg d$^{-1}$ (July 22-29). To avoid misleading, we have revised the definition of the OCR as the biochemically-induced oxygen consumption with time. A higher OCR value indicates stronger oxygen consumption and a negative value indicates oxygen production due to biochemical processes (e.g., photosynthesis) (Page 4, Lines 112-114 of our revised MS). We have also corrected the statement as "DO declined at a rate of ~ 9 $\mu$mol O$_2$ kg$^{-1}$ d$^{-1}$ from July 20-22, when the winds remained strong, and the OCR decreased to ~ 5.5 $\mu$mol O$_2$ kg$^{-1}$ d$^{-1}$ from Leg 2 to Leg 3 (from July 22-29)" in our revised manuscript (Page 19, Lines 452-454).

*L317: you mean horizontal diffusion?*

**[Response]**: The reviewer is correct, but we found that the horizontal diffusion for oxygen between the hypoxic zone and its surrounding environment might be much smaller than that

induced by lateral advection. We thus have revised it to solely discuss the lateral advection (Page 19, Lines 457-462 of our revised MS).

*L316-320: your analysis is not well supported, can you discuss your results and be more quantitative? If not, please do not extrapolate*

**[Response]**: We agree with the reviewer that the horizontal diffusion for oxygen was hypothetical and we have removed this discussion in our revised manuscript.

*L324: Figure 3 indicates the intrusion of warm and salty subsurface waters at station F303 after the typhoon, can you comment on that and how it fits with your analysis?*

**[Response]**: We appreciate the critical comment. Indeed, the time-series observations at station F303 indicated an upward intrusion of warm and salty subsurface waters (Fig. R1). However, this upward intrusion was progressively suppressed by the surface plume. The water column below the pycnocline became almost vertically well-mixed towards the end of the time-series observations before Leg 2, showing small vertical variabilities in profiles of temperature, salinity and DO concentrations. Therefore, this upward-intrusion before Leg 2 will not compromise our assumption for estimating the OCR from Leg 2 to Leg 3.

Specifically, we attributed the upward intrusion to the subdued vertical mixing due to weakened winds (Fig. R6c), resulting in a less well-mixed water column. The upward-intruded warm and salty waters (temperature > 28.1 °C and salinity > 32.2) resulted from the antecedent strong winds-induced vertical mixing upon disturbance by typhoon, which mixed the warm, brackish surface waters (> 29 °C; Fig. R8a) downward to increase the temperature of bottom waters. Indeed, the bottom waters were > 28.1 °C with water depths > 16 m (the depth of the bottom layer at station F303) during Leg 2 (Fig. R7). The relatively cooler surface waters might result from the heat loss at the air-sea interface due to the reduction in air temperature by ~ 2-3 °C during the typhoon period and afterwards (Fig. R10).

[Figure]

**Figure R10:** Air temperature at the Hong Kong Observatory in July 2018. The shaded area indicates the cruise periods for Leg 1 (grey), Leg 2 (pink) and Leg 3 (blue), respectively.

*L336: end of sentence: (Figure 5)*

**[Response]**: Accepted. We have added the reference of the figure at the end of the sentence (Page 20, Lines 468-470 of our revised MS).

*L338: those are really rough estimates. It is impossible to see what are the bottom $O_2$ values in Figure 2 so it is difficult to judge your reasoning. Is the OCR=-15 value an average over the sampling area? bottom $O_2$ values seem rather low over the entire area sampled during Leg 3*

**[Response]**: We agree with the reviewer that it was a first order estimate for upscaling to a larger area. This larger area implied the oxygen-deficient zone which likely developed into the hypoxic zone off the PRE. Therefore, we used the OCR of ~ 15 μmol $O_2$ kg$^{-1}$ d$^{-1}$ averaged over the oxygen-deficient zone (DO < 94 μmol kg$^{-1}$) during Leg 3, not over the whole sampling area. Despite low bottom oxygen values over the entire area during Leg 3, we showed relatively large spatial gradients of the total DO changes in bottom waters from Leg 2 to Leg 3 in Fig. R11 (Page 17, Figure 8 of our revised MS). The pattern of the OCR was almost consistent with the total DO changes, which is aligned well with the notion that mixing-induced DO changes were much smaller than the total DO changes. We have also improved the quality of figures by enlarging labels and boldening lines (Fig. R7; Page 11, Figure 4 of our revised MS) for better presentations.

[Figure]

**Figure R11:** Distributions of (a) total DO changes (ΔDO, μmol kg$^{-1}$) and (b) the biochemical-induced oxygen consumption rate (OCR, μmol $O_2$ kg$^{-1}$ d$^{-1}$) between Leg 3 and Leg 2 on the inner NSCS shelf off the PRE.

*L341: Figure 3 shows that this is more variable than assumed here*

**[Response]**: We agree with the reviewer that the time-series observations showed slightly more variable. This is because the time-series observations were conducted after the passage of the typhoon but still with relatively strong winds decreasing from ~ 10 m s$^{-1}$ to ~ 7 m s$^{-1}$ (Fig. R6c). The upward intrusion of warm and salty bottom waters was also progressively suppressed by

the freshwater input-induced stratification towards the end of the time-series observations (Fig. R1). This would not conflict with our assumption here for a common scenario during the late spring, typically with a monthly average wind speed of < 6 m s$^{-1}$. The subsurface waters below the pycnocline therefore could be assumed almost well-mixed.

*L345: how did you come up to the value 183 from your assumption? do you assume mixing rather than water replacement during intrusion?*

**[Response]**: Thanks for the critical comment. Based on our assumption, the well-oxygenated offshore surface waters have an DO concentration of ~ 194 μmol kg$^{-1}$. We also estimated that shoreward-intruded subsurface waters reduced the initial DO level by 8.6±1.7 % of the oxygen decline for hypoxia formation —— ~ 11 μmol kg$^{-1}$ when the oxygen level decreased from ~ 194 μmol kg$^{-1}$ to the threshold of hypoxia (63 μmol kg$^{-1}$). The initial DO level for the biochemical-induced oxygen consumption for hypoxia formation was therefore ~ 183 μmol kg$^{-1}$. Actually, we assumed water mass mixing instead of water replacement during the intrusion (Page 20, Lines 475-479 of our revised MS).

*L346: This does not match your calculation above with the average OCR, why are you assuming the maximum OCR during spring?*

**[Response]**: We appreciate the critical comment. We used the maximum OCR during late spring to estimate a minimum time for the occurrence of hypoxia hotspots after the initiation of significant oxygen depletion. If the hypoxic zone developed to a larger spatial coverage, the time would be longer than the minimum time, which was consistent with our above estimates for the reinstatement of hypoxia in the summer of 2018.

*L347: you should compare your values with similar systems, i.e. river-dominated estuaries (8-89 days). Also you could discuss your estimates in comparison of the PRE values provided in the reference.*

**[Response]**: Accepted. We have compared our OCR estimates with previous studies in this study area and other large river-dominated shelf systems: "This result is larger than that estimated by Fennel and Testa (2019) — 4 days — using the modelled OCR of ~ 34 μmol O$_2$ kg$^{-1}$ d$^{-1}$ in the water column with a sediment oxygen demand of ~ 2.1 g m$^{-2}$ d$^{-1}$, which were only applicable to the hypoxia formation in the Lingdingyang sub-estuary with shallower waters (~ 5 m) (Zhang and Li 2010). This result is still at the lower end of the hypoxia formation timescale in large river-dominated shelves globally (e.g., East China Sea off the Changjiang estuary, Northern Gulf of Mexico and Northwestern Black Sea), which varies from 8 to 89

days for hypoxia to develop once initiated (Fennel and Testa 2019). This short hypoxia formation timescale likely owes to a high OCR in relatively warm subsurface waters fuelled by abundant labile organic matters (Su et al., 2017; Zhao et al., 2020)" in our revised manuscript (Page 20, Lines 481-488).

*L368-403: The last 2 paragraphs are not related to your results*

**[Response]**: Accepted. We have revised these paragraphs by (1) checking the precipitation and river discharge after the typhoon, (2) correlating our discussion closely with our observtions of stronger post-storm blooms and their offshore advection along with the river plume, and (3) estimating changes in the frequency and intensity of tropical cyclones and the time interval between two successive tropical cyclones during the wet seasons. These revisions aim to discuss about the evolution of hypoxia upon disturbance by tropical cylones and its response to changes in the frequency and intensity of tropical cyclone activities (Pages 21-23, Lines 512-551 of our revised MS).

*L368: You did not mention/discussed the phytoplankton bloom in leg 2*

**[Response]**: As responded above, we described the strong post-storm blooms in Section 3.2 — Destruction of hypoxia by typhoon. We have further discussed the response of the post-storm hypoxia development to tropical cyclone activities, which might largely depend on the dynamics of the strong post-storm blooms: "Enhanced vertical mixing and/or freshwater discharge supplied large amounts of nutrients to the surface layer to fuel phytoplankton blooms following large storms (Zhao et al. 2009, Ni et al. 2016, Wang et al. 2017), as shown in Fig. 3 that strong blooms occurred in the surface plume along the coast with much higher Chl *a* concentrations during Leg 2 than that during Leg 1. The fresh autochthonous organic matter, together with the resuspended sedimentary organic carbon, provides sufficient substrates for microbial respiration in a re-stratified water column, leading to renewed or even exacerbated bottom water oxygen depletion (Zhou et al. 2012, Song et al. 2020)" (Page 21, Lines 518-523 of our revised MS).

*L379: why?*

**[Response]**: The statement was confusing and hypothetical. We have revised the statement based on our observations — "Whether it can develop into more severe hypoxia compared to that found initially during Leg 1 depends on the net OCR and water column stability, up until the passage of the next storm, Typhoon BEBINCA (Fig. 1d)" (Pages 21-22, Lines 527-529 of our revised MS) — bacause the post-storm blooms advected offshore might reduce the

downward transport of labile organic matters to fuel the oxygen depletion in the subsurface waters.

*Figure 6: I am not sure that annual averages are very pertinent, an average value per event might be more useful. For wind direction this could be presented as a pie chart. For wind speed, the time series is not very informative, may be think about an other way of presenting the results. This is somewhat included in Figure 6a but it would be interesting to know what is the maximum time between wind events for each year (or have some statistics based on your estimate of OCR). Also remember that hypoxia did not occur for most of the period shown here. panel c: please see comment regarding wind vectors in Figure 1, make sure those are right*

**[Response]**: We appreciate the critical comments and constructive suggestions. The annual average was actually the annual average of the local maximum wind velocity for each tropical cyclone in the NSCS. We have revised the figure to additionally show the maximum wind speed of each tropical cyclone (Fig. R4b; Page 22, Figure 9 of our revised MS). For wind direction, we re-plotted it as roses of winds with the classifications of storm intensities and the duration times of tropical cyclones in the northern South China Sea (NSCS) (Fig. R4d, e). We also have calculated the time interval between two successive tropical cyclones for each year (Fig. R4c). In our calculations and statistics on historical tropical cyclones, only tropical cyclones that impacted the NSCS from May to September were taken into account because hypoxia often occurred from late spring to summer and disappeared in autumn in this study area.

*L398: it depends on the direction, offshore intrusions would presumably bring lower $O_2$ waters*

**[Response]**: We agree with the reviewer that offshore intrusions would bring lower-oxygen waters due to the deoxygenation of oceanic waters and it might contribute to the oxygen loss and the formation of hypoxia more significantly, but we have removed this statement as it was less related to our topic.

*L409: "lowest ever recorded"*

**[Response]**: Accepted. We have corrected the statement as "Eutrophication-induced hypoxia off the PRE was exacerbated with an enlarged area of ~ 660 $km^2$ and the lowest ever recorded regional DO concentration of 3.5 $\mu mol\ kg^{-1}$ (~ 0.1 mg $L^{-1}$)" in our revised manuscript (Page 23, Lines 558-559).

*L417: This is speculation, higher discharge may lead to lower nutrient concentrations in river waters, more export to deep areas where hypoxia does not occur*

**[Response]**: We agree with the reviewer that the higher discharge may decrease nutrient concentrations in river waters, but the total discharge of nutrients may increase due to strong flushing (Guo et al., 2008). In this study, stronger blooms occurred during Leg 2 than that during Leg 1, significantly utilizing the nutrients and producing organic matters to fuel the oxygen depletion in the subsurface waters. However, we observed an offshore shift of the blooms along with the river plume during Leg 3. Therefore, whether more nutrients will be exported to deep areas also depends on the phytoplankton uptake, the wind direction and alongshore currents that drive the river plume to spread offshore and their dominance on the water residence time. We have revised this statement without nutrient loading in our revised manuscript (Page 23, Lines 548-550).

**Minor comments/edits:**

*L80: can you provide the number of stations in parenthesis for each leg, it is difficult to estimate it from Figure 1b*

**[Response]**: Accepted. We have added the number of stations for each leg as "Almost all stations in Leg 1 (56 stations) were revisited during Leg 2 (56 stations, including 4 stations differing from Leg 1), and nearly half again during Leg 3 (27 stations)" in our revised manuscript (Page 3, Lines 79-80).

*L81: suggestion: "on the way back to port"*

**[Response]**: Accepted. We have revised the statement as "Eight stations were additionally revisited on the way back to the port on July 31" in our revised manuscript (Page 3, Lines 80-81).

*L155-164: You are mixing results and discussion*

**[Response]**: Accepted. We have re-organized the paper by combining results and discussions to a clear narrative with an outline listed above in our response to the general comment. Therefore, we kept them and will further compare the results with that in the summer of 2014 (Su et al., 2017) (Page 8, Lines 209-214 of our revised MS).

*Figure 2: there is no point showing the river labels, they are way too small. Also the contour labels cannot be seen and the color bars are way too small. I suggest you move the colorbar to the top of each column and make it thicker with larger fonts An alternative suggestion is to*

*split the figure into surface and bottom figures and flip the rows to columns to make larger panels*

**[Response]**: Accepted. Accordingly, we have added the river label in Fig. R6a (Page 5, Figure 1 of our revised MS) and separated this figure into two figures showing the surface and bottom distributions, respectively (Fig. R7 and Fig. R8; Page 10, Figure 3 and Page 11, Figure 4 of our revised MS). We also have enlarged the panels and contour labels for all figures.

*Figure 4: The labels and lines are very small*

**[Response]**: Accepted. We have enlarged the labels and boldened the lines (Fig. R3; Page 15, Figure 7 of our revised MS).

*L312: not clear, the sentence should be rephrased*

**[Response]**: Accepted. We rephrased the sentence as "During the hypoxia formation, the DO concentrations are in a non-steady state as oxygen sinks exceed sources. To shift towards a balance between oxygen consumption and replenishment for the maintenance of hypoxia, the OCR might decrease or the physical-induced oxygen supply increases" (Page 19, Lines 444-448 of our revised MS).

**Anonymous Referee #2**

*I found this manuscript to be a useful contribution to our understanding of the spatial and temporal nature of oxygen depletion as a large coastal system responds to large events. The narrative is relatively easy to follow and the results are clearly communicated with figures. I think the analysis could benefit from a small amount of additional computations, but I also think that the results and discussion section needs to be reorganized. There is substantial mixing of results and discussions between the two sections, and I think it would be best and easiest to simply combine the two sections into one "Results and Discussion" section that is reorganized into a clear narrative.*

**[Response]**: We are grateful that the reviewer valued our study. We also appreciate the critical comments and constructive suggestions from the reviewer, which will be fully considered in our revised manuscript. In brief, we have followed the suggestions to reorganize the paper by combining results and discussion into a clear narrative (Pages 8-23 of our revised MS), with an outline as below:

3 Evolution of intermittent hypoxia off the PRE

        3.1 Extensive hypoxia before the typhoon

        3.2 Destruction of the hypoxia by the typhoon

        3.3 Reinstatement of the hypoxia after the typhoon

4 Maintenance, destruction and reinstatement of coastal hypoxia

        4.1 Water column stability

        4.2 Oxygen sinks and hypoxia formation timescale

            4.2.1 Mixing-induced oxygen sinks

            4.2.2 Biochemical-induced oxygen sinks

            4.2.3 Hypoxia formation timescale

        4.3 Imprint of tropical cyclones on the evolution of coastal hypoxia

We additionally have (1) calculated the vertical diffusion for oxygen using the density-based eddy diffusivity from Cui et al. (2019) with our observed DO concentrations and estimated buoyancy frequency (Page 14, Lines 306-308 of our revised MS); (2) calculated and plotted the surface-to-bottom salinity and temperature differences to show the spatial distribution of stratification (Page 13, Lines 292-296 of our revised MS); (3) calculated the potential maximum surface area of the hypoxia associated with tidal fluctuations. The spring-to-neap tidal oscillations lead to variations in the DO concentration off the Lingdingyang sub-estuary with a maximum neighboring oxygen range of 0.5 mg L$^{-1}$ (Cui et al., 2019). Assuming the

observed DO concentration in Leg 1 (from a neap tide to a spring tide; Fig. R6e) was overestimated by 0.5 mg L$^{-1}$ (i.e., ~ 15 μmol kg$^{-1}$), the total area of the hypoxic and oxygen-deficient zone would be at most ~ 990 km$^2$ and ~ 1930 km$^2$, respectively, 34-50% larger than our observed areas (Page 16, Lines 348-356 of our revised MS); and (4) discussed our results and estimates by comparing with previous studies in this study area and other large river-dominated shelf systems (Page 18, Lines 427-429 and Page 20, Lines 481-488 of our revised MS). We will address these concerns from the reviewer in our responses as of below.

Below are some specific and more general comments for the authors to consider:

*(1) Line 49: "typhoons" should be plural*

**[Response]**: Accepted. We have corrected it in our revisions (Page 2, Line 48 of our revised MS).

*(2) Figure 1: It is a little difficult to discern the station locations of the different legs, given the overlapping in the circles. One suggestion could be to use different symbols to present (1) stations visited on all legs, (2) stations visited on legs 1+2, (3) stations visited on legs 2+3, and (4) stations visited on 1+3.*

**[Response]**: We appreciate the suggestion. We have modified Figure 1b to show the stations more clearly: (1) stations visited in all legs, (2) stations visited only in Leg 1 and Leg 2, (3) stations visited only in Leg 1, and (4) stations visited only in Leg 2 (Fig. R1b) (Page 5, Figure 1 of our revised MS).

*(3) Although Figure 3 nicely illustrates how stratification returned after the cyclone, it does not capture any patterns over space and it does not capture the entire coverage of the study in time. Figure 2 provides a nice, qualitative picture of the changes in water properties over time and space, but I think it might be helpful to also generate maps of the stratification changes, perhaps by plotting max N2 over space or the difference in temperature and salinity (or density) between surface and bottom waters. This would have the benefit of showing if stratification was weaker after it was reinstated than before the typhoon, where stratification was strongest, and how it related in space to hypoxia.*

**[Response]**: We agree with the reviewer that changes in spatial distribution of stratification would help better understand how it is related to hypoxia in space. Following suggestions, we have plotted the difference in temperature and salinity between surface and bottom layers (Fig. R12; Page 14, Figure 6 of our revised MS). We also have revised our discussion on the effect

of stratification on the formation and maintenance of hypoxia: "The surface-to-bottom salinity difference showed large values within the surface plume area, which almost covered the bottom hypoxic zones. Exceptions only occurred to the hypoxia zone off the Modaomen sub-estuary, where the surface-to-bottom salinity differences were relatively small but the temperature differences were large (i.e., $\Delta T_{b-s} < -4$ °C) due to the shoreward intrusion of cold offshore subsurface waters. The regions occupied by the surface plume and the shoreward-intruded shelf bottom waters therefore overlapped, resulting in a more stable water column where a patchy hypoxic zone could persist for more than 5 days (Cui et al., 2019)" (Page 13, Lines 292-298 of our revised MS).

[Figure]

**Figure R12**: Surface-to-bottom temperature (a-c) and salinity (d-e) distributions off the PRE during Leg 1 pre-typhoon, and during Legs 2 and 3 post-typhoon. $\Delta T_{b-s}$ and $\Delta S_{b-s}$ represent the difference in temperature and salinity between the bottom and surface layer, respectively.

*(4) Figure 4 – I think it would be more interesting to also show vertical oxygen distributions on figure 4, to show where hypoxia exists relative to the vertical structure and stratification.*
**[Response]**: Accepted. We have added oxygen profiles at multiple stations accompanying the vertical distributions of temperature, salinity and buoyancy frequency (Fig. R3; Page 15, Figure 7 of our revised MS). The DO concentrations decreased sharply at the base of the surface plume (salinity ~ 30).

*(5) You report on the decline in oxygen concentration in the water column after the typhoon passing as a metric of oxygen consumption rate. It would make the paper more compelling,*

*and help the discussion, to compare these rates of oxygen depletion to similar rates published in other systems (e.g., Testa and Kemp 2014, others?)*

**[Response]**: We have added such comparisons in our revisions: "Our estimated OCR is comparable in magnitude to the community/bacterial respiration rate from previous studies in this study area (9.6 μmol $O_2$ $kg^{-1}$ $d^{-1}$, Su et al., (2017); 7.9 to 19.0 μmol $O_2$ $kg^{-1}$ $d^{-1}$, Cui et al., (2019); 16.8±8.9 μmol $O_2$ $kg^{-1}$ $d^{-1}$, Li et al., (2019)) and within the range in other estuaries and coastal systems (Dortch et al., 1994; Robinson 2008)" (Page 18, Lines 427-429 of our revised MS).

*(6) Line 315-319: Can you estimate the oxygen diffusivity rate from your data, based on any published estimates of diffusivity for the region, or estimated from your density profiles? This would allow you to be more quantitative in your comparison of OCR and diffusivity as eventually balancing. I think you could also speculate, perhaps with data, why OCR could have possibly declined, either as the post-bloom organic material was exhausted or due to oxygen limitation of respiratory uptake?*

**[Response]**: Accepted. We have estimated the vertical diffusion for oxygen based on the published diffusivity from Cui et al., (2019) along with our observed DO concentrations and estimated buoyant frequency: "Using the density-based eddy diffusivity ($K_z$) of $< 5 \times 10^{-6}$ $m^2$ $s^{-1}$ for $N^2$ larger than $1 \times 10^{-3}$ (Cui et al., 2019), we estimated the vertical DO diffusitivity (VDIF $= K_z \times (\partial DO / \partial z)$) of $\sim 0.25$ g $m^{-2}$ $d^{-1}$ with a maximum of 0.54 g $m^{-2}$ $d^{-1}$ in the top 10 m at stations A8 and A11, which was comparable to the results from Cui et al. (2019). It therefore acted as a barrier layer, with weak dissipation of oxygen into the subsurface waters" (Page 14, Lines 306-309 of our revised MS). As inhibited by freshwater input-induced stratification, the oxygen supply by the vertical diffusion was much smaller than the biochemical consumption, leading to a relatively strong net oxygen consumption.

The decline in OCR could result from a reduced supply of labile organic matter or oxygen limitation of respiratory uptake. Oxygen enriched incubations of unfiltered water samples revealed that the OCR could be significantly enhanced when the initial *in situ* DO concentration was low (e.g., $\sim 30$ μmol $kg^{-1}$), but changed little when the *in situ* DO concentration was higher than $\sim 90$ μmol $kg^{-1}$ (He et al., 2014). Here, the bottom DO concentrations were $\sim 180$ μmol $kg^{-1}$ at station F303 in the time-series observations, much higher than the oxygen threshold for respiratory uptake. However, the strong post-storm blooms shifted offshore with the river

plume and the Chl *a* concentrations over the hypoxic zone also decreased in Leg 3 as comapred to Leg 2 (Fig. R7), reducing the downward transport of labile organic matter to the hypoxic zone. The decline in the OCR in this study thus very likely owed to a reduced supply of labile organic matter (Page 19, Lines 448-457 of our revised MS).

*(7) Paragraph on Line 389: This paragraph reads more like an essay on the factors driving hypoxia and vulnerable to climate change, and does not really discuss the specific details of this study. I suggest deleting it, perhaps keeping the cyclone points for the prior paragraph on cyclone effects.*

**[Response]**: Accepted. We have removed the discussion on the exacerbation of hypoxia under a changing climate, but keep discussions on the response of coastal hypoxia to changes in the frequency and intensity of tropical cyclone activities. Statistics on the local maximum wind speed and wind direction of tropical cyclones and the time interval between two successive tropical cyclones have been added based on the historical dataset of tropical cyclones that impacted the northern South China Sea from May to September during the period of 1975-2019 (Table R1; Fig. R4) (Page 23, Lines 541-551 of our revised MS).

*(8) I think you should combine the Results and Discussion Sections into one, well-organized narrative. As it stands, there are multiple places where results are reported in the discussion, or there are even methods in the discussion. This would allow you to more clearly and sequentially tell the story of your study. Below are some specific examples to guide this effort: (a) Line 225-239 is largely results and even methods, but is included in the discussion without substantial discussion of the results in the context of the study. (b) Line 285-290. Here, you are describing the method you already described. Move to methods and remove redundancy. (c) Paragraphs beginning on lines 332 and 343 can be combined*

**[Response]**: Following suggestions, we have moved all method descriptions to the section of Materials and methods (Page 4, Lines 102-109 and Pages 6-7, Lines 155-177 of our revised MS) and reorganized the paper by combining results and discussion into a clear narrative, with an outline as listed above in our response to the general comment.

**References**

[revised manuscript text omitted]

---

## Author Response (AR2)

*Re: Destruction and reinstatement of coastal hypoxia in the South China Sea off the Pearl River Estuary" by Yangyang Zhao et al.*

March 07, 2021

Dear Editor,

Thank you for your time in handling our paper. Enclosed please find our revised MS entitled "Destruction and reinstatement of coastal hypoxia in the South China Sea off the Pearl River Estuary" by Yangyang Zhao et al.

During the revisions, we have fully considered the comments and suggestions from the reviewers. Briefly, we elaborated the shoreward intrusion of deep shelf waters originating from the coastal current which flows northeastwardly in summer. Following suggestions, we have rephrased or removed statements that were not supported by our observations.

Finally, we would like to take this opportunity to thank the reviewers for their constructive comments and suggestions, which significantly improved the quality of the paper. We sincerely hope that our revision will meet the standards of *Biogeosciences*.

Sincerely,

Minhan Dai
Corresponding author
State Key Laboratory of Marine Environmental Science
Xiamen University
Xiamen 361005, China
Phone: 86-592-218-2132
Fax: 86-592-218-0655
E-mail: mdai@xmu.edu.cn

**Anonymous Referee #1**

**General comments:**

The revisions significantly improved Zhao et al manuscript. The choice of mixing Results and Discussion improves the readability. The drawback is that some of the statements are not supported by observations. This was already the case in the previous version of the manuscript. I encourage the authors to read through their manuscript and remove or rephrase some of their claims that are not supported by their observations. Specific examples are listed in the comments below. For example I would remove the section/paragraph about tidal mixing and future state that are not linked to the current results.

[Response]: We appreciate the constructive comments and suggestions from the reviewer. We have further revised the MS accordingly as explained in our responses as of below.

**Specific comments**

L168: Do you have supporting observations for this assumption?

[Response]: Yes. This assumption is based on our samples collected at depths of 43-50 m along the ~ 50-m isobaths off the Pearl River Estuary in July 2017. The DO concentrations of these samples were averaged ~180.9 μmol kg$^{-1}$, nearly 16% of oxygen deficit relative to the saturation level for the water mass with a temperature of 22.5°C and a salinity of 34.5, i.e., the upwelled subsurface water of which the endmember values were adopted from Zhao et al. (2020).

L199-200: Mixing Results and Discussion doesn't mean you can extrapolate your results without supporting observations. The wording is important. In this sentence a better wording could be: "The freshwater bulge also featured a relatively weak bloom, with Chl a concentrations of ~10 μg L$^{-1}$ and DO of ~250 μmol kg$^{-1}$ (equivalent to a DO saturation level of ~ 125 %). Similar blooms were previously associated with high nutrient concentrations and a long water residence time in the plume (Lu and Gan 2015)."

[Response]: We appreciate the reviewer's comment. Following suggesions, we have revised the sentence as "The freshwater bulge also featured a relatively weak bloom, with Chl $a$ concentrations of ~ 10 μg L$^{-1}$ and DO of ~ 250 μmol kg$^{-1}$ (equivalent to a DO saturation level of ~ 125 %). Similar blooms were previously observed to be associated with high nutrient concentrations, a sufficently long water residence time and an abundance of photosynthetically active radiation (Lu and Gan, 2015)." (Page 8, Lines 204-207 of our revised MS).

L201-202: That does not seem like a relevant explanation for the lower water temperature. 1) there is no indication that the lower surface temperature is the direct result of the air temperature, 2) air temperature is lower only at the end of the leg for a short period of time and 2) air temperature could be the same in the other areas.

"during data collection", "at the time of sampling" would be better than "visits".

[Response]: We appreciate the reviewer's comment. We plotted the surface water temperature measured at a depth of 1 m for all stations along the cruise track with the air temperature (Figure R1), showing a consistent trend of changes in temperature. Specifically, the air temperature was lower at the end of Leg 1, also in consistence with our observations when we sampled stations to the southwest of Hong Kong. Following suggestions, we revised Figure S1 (Page 2 of our revised Supplement) by adding the surface water temperature and the statement as "Exceptions occurred to the southwest of Hong Kong where the air temperature was 2-3 °C lower at the time of sampling" (Page 8, Lines 208-209 of our revised MS).

[Figure]

**Figure R1:** Surface water temperature and air temperature in July 2018. Surface water temperature was measured at a depth of 1 m for all stations along the cruise track. Air temperature was recorded at the Hong Kong Observatory. The shaded area indicates the cruise periods for Leg 1 (grey), Leg 2 (pink) and Leg 3 (blue), respectively.

L204: Bottom shelf waters are found in z>10-20m below plume waters. They are not "intruded". The plume is attached to the bottom in the inshore area (0<z<20) according to Figure 4.

[Response]: Here "shelf benthic waters" actually indicate deep shelf waters originated from the coastal current over the middle/outer shelf which flows northeastwardly in summer. The coastal current is advected upslope towards the coastal zone off the Pearl River Estuary due to irregular topography (Gan et al., 2009; Liu et al., 2020). This upslope transport prevents the buoyant plume from being bottom-advected over the shelf deeper than 20 m, and, in response to the eastwardly strengthened coastal current, the onshore invasion of deep shelf waters

strengthened eastwards (Liu and Gan, 2020). Therefore, we have replaced "shelf benthic waters" by "deep shelf waters" to avoid misleading.

L208-209: please rephrase "a region... measurements"

**[Response]**: Accepted. We have revised the statement as "Additionally, a smaller-scale hypoxic zone appeared beneath the surface bloom near the Huangmaohai sub-estuary, a region where hypoxia was also ever reported but not fully covered by survey measurements." (Page 8, Lines 214-216 of our revised MS).

L212-213: how did you calculate these areas?

**[Response]**: We interpolated DO concentrations into 0.5'×0.5' grids over our sampling regions with observed DO data using the Kriging interpolation method. For grids with DO < 63 or 94 $\mu$mol kg$^{-1}$, we integrated their spatial area for the hypoxic or oxygen-deficient zones, respectively.

L215-216: please remove this statement. You cannot draw long term conclusions by comparing 2014 and 2018. Also, the sampling area was different.

**[Response]**: Accepted. We have removed this statement accordingly.

L224-225: That is not supported by your observations from Fig S1. Air temperature at the Hong Kong observatory was >28 at this time.

**[Response]**: During the time-series observations from July 19-20, the air temperature at the Hong Kong observatory was ~ 28 °C with a range of 26-31.6 °C (Figure R1).

L229-230: Same comment here, be careful with the wording. The bloom is within the plume and therefore more likely associated with the riverine nutrient input.

**[Response]**: We appreciate the reviewer's comment. The bloom was indeed within the plume, and the plume waters were a mixture of freshwater and seawater, the latter of which had an elevated nutrient concentration due to the strong vertical mixing during the typhoon period. This relatively high nutrient seawater increased nutrient supplies and fueled the post-storm bloom. Therefore, the post-bloom was potentially fuelled by the nutrients mixed upward from the depth in addition to riverine inputs.

L239-241: "Although... pycnocline": This is not yet supported, please remove.

**[Response]**: During Leg 2, the similar distributions of salinity and temperature at middle and bottom layers (Figure R3, R4) indicate that the subsurface water column remained relatively well-mixed, whilst the discrepancy between the surface layer and the middle/bottom layers (Figure R2, R3, and R4) indicates the revitalization of density stratification due to freshwater buoyancy and weakened winds (Figure R5c). The following sentence also supports that the oxygen decline after the typhoon. Therefore, we would like to keep this statement — "Although the water column remained relatively well-mixed in the subsurface layer, freshwater buoyancy and weakened winds facilitated the revitalization of density stratification and subsequent oxygen decline below the pycnocline" (Page 9, Lines 248-250 of our revised MS).

L249-250: provide supporting observations or remove this statement

**[Response]**: Figure R2c shows that the surface temperature was warmed to over 30 °C during Leg 3, increasing the vertical thermal gradient relative to that during Leg 2 (Figure R2, R3, R4). The increased vertical thermal gradient strengthened the density stratification which was thus mainly driven by the freshwater inputs. We have revised the statement as "The surface layer was warmed up to over 30 °C (Fig. 3c), increasing vertical thermal gradients relative to Leg 2 (Fig. 3 and Fig. 4) and strengthening the stratification (Allahdadi and Li, 2017)" (Page 9, Lines 258-260 of our revised MS).

L265: What is the depth of this subsurface (mid-) layer shown in Fig. S2?

**[Response]**: The depth of the middle layer shown in Fig. S2 depends on the geographical locations of the stations, varying from 3-5 m at nearshoremost stations to ~ 15 m at offshoremost stations.

L265-266: "likely... winds". again, this is not shown

**[Response]**: Figure R5d shows that the southeasterly winds dominated Leg 3, which constrained the plume to the coast and even to penetrate into the subsurface layer (Huang et al., 2019; Li et al., 2021), as shown in Figure R3 and R4. We have revised the statement as "Similar to Leg 1 and Leg 2, the surface waters penetrated into the subsurface layer along the coast (Fig. 4f), likely forced by the downwelling-favourable winds (Fig. 1d) …" (Page 12, Lines 276-278 of our revised MS).

[Figure]

**Figure R2:** Distributions of temperature (°C), salinity, DO (µmol kg⁻¹) and Chl *a* concentrations (µg L⁻¹) at the surface water layer off the PRE during Leg 1 pre-typhoon, and during Legs 2 and 3 post-typhoon. The white and magenta contours in (g) and (w) show the hypoxic (DO < 63 µmol kg⁻¹) and oxygen-deficit (DO < 94 µmol kg⁻¹) zones. Figures were produced using Ocean Data View v. 5.3.0 (http://odv.awi.de, last access: 08 June 2020)

[Figure]

**Figure R3:** Distributions of temperature (°C), salinity, DO (μmol kg⁻¹) and Chl *a* concentrations (μg L⁻¹) at the middle layer off the PRE during Leg 1 prior to Typhoon, and during Legs 2 and 3 post-typhoon. The almost homogeneous spatial distribution was similar to the bottom layer after being disturbed by the typhoon, and reaeration along the coast in shallow waters was forced by easterly winds in Legs 1 and 3 when hypoxia developed.

[Figure]

**Figure R4:** Distributions of temperature (°C), salinity, DO (μmol kg⁻¹) and Chl *a* concentrations (μg L⁻¹) at the bottom water layer off the PRE during Leg 1 pre-typhoon, and during Legs 2 and 3 post-typhoon. The white and magenta contours in (g) and (w) show the hypoxic (DO < 63 μmol kg⁻¹) and oxygen-deficit (DO < 94 μmol kg⁻¹) zones.

[Figure]

**Figure R5**: (a) Map of the study area on the shelf of the northern South China Sea (NSCS), showing the track of Typhoon SONTIHN (circles) across the NSCS during July 16-24, 2018. The color of the circles represents the magnitude of wind speed. Additionally, the smaller circles denote tropical depression (wind speeds ≤ 17.1 m s$^{-1}$) and the larger circles denote tropical storm (wind speeds within 17.2-32.6 m s$^{-1}$). The arrows denote the locations of the typhoon as marked with time and wind speed. The grey lines are the depth contours at 50 and 200 m. (b) Sampling stations on the NSCS shelf off the Pearl River Estuary in summer 2018. The pink, green, purple and orange circles denote the stations surveyed in all three legs, only both Leg 1 and Leg 2, only Leg 1 and only Leg 2, respectively. Time-series observations were conducted at Station F303 as marked by the star, and vertically high-resolution samplings were conducted at stations marked with bold circles. (c) The wind speed and (d) wind direction at Waglan Island (triangle in (b)) from May to August, 2018. Bars at the bottom of (d) mark times when tropical cyclones impacted the NSCS. (e) The tidal height at the Dawanshan gauge station near Station F303 from May to August, 2018. The shaded area indicates the cruise periods for Leg 1 (grey), Leg 2 (pink) and Leg 3 (blue), respectively.

L272: "expanded... isobath": where is this shown?

[Response]: We have removed this statement — "…and expanded along the 20-m isobaths".

L274-277: Again, this is not supported by your observation. It can be discussed (as in a regular discussion) but you cannot make this type of conclusion based on your observations.

[Response]: Comparing the bottom distributions of temperature and salinity during Leg 2 and Leg 3 (Figure R4), we found that the low-temperature, high-salinity bottom waters to the southwest of Hong Kong were upslope-invaded deep shelf waters in response to topographical-driven upwelling (Gan et al., 2009; Liu et al., 2020). By further comparing the environment settings associated with the two hypoxic centers, off the Modaomen sub-estuary and to the southwest of Hong Kong, we contend that "the shoreward intrusion of deep shelf waters is not a prerequisite for the initiation of hypoxia formation off the PRE, but it contributes to the reinstatement of hypoxia southwest off Hong Kong" (Page 12, Lines 286-288 of our revised MS).

L282: This section title is a bit redundant with the previous subsection titles. In part 3 you look at the temporal evolution of hypoxia (as stated in the section/subsection headers) and in part 4 you look into the mechanisms that lead to the temporal evolution observed in part 3. This should be reflected in the section headers.

[Response]: We appreciate the reviewer's comment. We have revised the section title of part 4 as "Physical and biochemcial controls on the evolution of intermittent hypoxia" (Page 13, Line 293 of our revised MS).

L292: The time series is only 26h long so you shouldn't draw too many conclusions. The end of the time series indicate that the location of the plume varies rapidly in space/time.

[Response]: Here we described the phenomenon based on our observations. Figure R6 clearly shows that low-salinity freshwater came along with high Chl *a* and DO concentrations at the surface layer in the second half of the observations. The signal of oxygen decline beneath the freshwater was also observed.

[Figure]

**Figure R6 :** Time-series observations of (a) temperature (°C), (b) salinity, (c) DO (µmol kg$^{-1}$) and (d) Chl *a* concentrations (µg L$^{-1}$) at Station F303 (see Fig. 1b) from July 19-20, 2018 after the typhoon passage, showing the complete destruction and the subsequent rapid development of stratification.

L292-298: all hypoxic centers occur in cold/salty waters during leg 1.

The terminology "intrusion" is misleading as it tend to indicate that offshore bottom waters are advected onshore, whereas it seems that it is the bottom-attached plume that prevents offshore bottom waters to reach the z<20m area.

**[Response]**: As responded above, deep shelf waters were actually advected onshore in the area with depths > 20 m, as the bottom waters originate from the coastal current over the middle/outer shelf which flows northeastwardly in summer. This "intrusion" of deep shelf waters does not conflict with that bottom-attached plume prevents offshore bottom waters to reach the area with depth < 20 m. We have replaced "shelf benthic waters" by "deep shelf waters".

L317-340: You could shorten this paragraph to avoid redundancy with section 3.

[**Response**]: This paragraph focused on discussing the effect of wind speed and direction on the evolution of hypoxia, differing from the description in section 3. Some observations were referred to support our discussion. Following suggestions, we have shortened this paragraph by removing unnecessary descriptions to avoid redundancy with section 3 (Pages 14-15, Lines 328-348 of our revised MS).

L337: "with a limited spatial extent beneath the surface plume": What do you mean? vertical extent?

[**Response**]: This means that the surface plume overlapped with the bottom hypoxia with a limited horizontal extent due to an offshore or westward shift of hypoxic zones.

L345-360: You speculate here, this paragraph should be reduced to one short sentence where you mention that tide could influence your estimates of hypoxia extent.

[**Response**]: We have shortened the paragraph following suggesions. We are now focusing on effects of the spring-neap tidal oscillations (Page 8, Lines 221-225 of our revised MS).

L362-370: this paragraph is not necessary, you should focus on your results. In general, it is better to start a section by stating your results and then develop.

[**Response**]: We have removed this paragraph and started Section 4.1 by our observed mixing-induced oxygen sinks.

L371: "On the condition of a precedent restoration of density stratification": can you rephrase for clarity?

[**Response**]: This statement means that the density stratification was restored before the oxygen decline from Leg 2 to Leg 3. The restoration of density stratificaiton is also a prepresiquite for oxygen decline. We have revised the statement as "With the restoration of density stratification from Leg 2 to Leg 3,…" (Page 16, Line 356 of our revised MS).

L385-394: The main center of hypoxia off Modaomen occurs where there is no $O_2$ mixing (Figure 8d), but you seem to suggest that this area is influenced by the upwelled waters SW of Hong Kong. The two centers of hypoxia (off Modaomen and off Hong Kong) seem to have different dynamics.

[**Response**]: Although little physical oxygen mixing contributed to the hypoxia formation off Modaomen sub-estuary during Leg 3, the coverage of cold, high-salinity deep shelf waters over

this hypoxic zone during Leg 1 (Figure R4) indicates that the hypoxia formation during Leg 1 was influenced by deep shelf waters, which is identical to the upwelled waters SW of Hong Kong.

L405: what do you mean here?

**[Response]**: As deep shelf waters originate from the alongshore shelf current in the low-latitude, high-temperature oligotrophic NSCS and the NSCS shelf is narrower than the East China Sea shelf, they spend a shorter time intruding toward the coastal zone after diverting its direction from the continental slope. We have revised the sentence as "Comparing with these systems, the source water originating from the low-latitude, high-temperature oligotrophic NSCS (Wong et al., 2007) has a higher DO level (AOU ~ 35 μmol kg$^{-1}$). As the NSCS shelf is narrower than the East China Sea shelf, the source water spends a shorter time intruding toward the coastal zone after diverting its direction from the continental slope (Gan et al., 2009; Wang et al., 2014). These might explain the relatively low contribution of coastal upwelling to the oxygen depletion on the inner NSCS shelf off the PRE." (Page 17, Line 381-386 of our revised MS).

L416-418: please rephrase, this was not observed in your study.

**[Response]**: We have revised the statement as "The geographical locations of high-OCR zones at the bottom layer (OCR > 10 μmol O$_2$ kg$^{-1}$ d$^{-1}$) were not fully overlapped with those of surface blooms (Fig. 3l and Fig. 8c)" (Page 17, Lines 394-396 of our revised MS) and removed the statement "even though eutrophication-produced organic matters were primarily responsible for fuelling oxygen depletion in the hypoxic zone (Su et al., 2017; Yu et al., 2020; Zhao et al., 2020)".

L421: "only the western hypoxic centre was located"

This is linked to the comment above (L385-394): it suggests different mechanisms for hypoxia formation between the two hypoxia centers?

**[Response]**: Accepted. We have revised the statement as "In fact, only the west hypoxic centre was located beneath the surface bloom from Leg 2 to Leg 3" (Page 17, Line 399 of our revised MS). As reponsed above, deep shelf waters covered the two hypoxic centres, despite organic matter of different sources and/or forms fueling oxygen decline for hypoxia formation between the two hypoxic centres.

[Figure]

**Figure R7:** Distributions of (a) total DO changes ($\Delta DO$, μmol kg$^{-1}$), (b) mixing-induced DO changes ($\Delta DO^{mix}$, μmol kg$^{-1}$) and (c) the biochemical-induced oxygen consumption rate (OCR, μmol O$_2$ kg$^{-1}$ d$^{-1}$) between Leg 3 and Leg 2 on the inner NSCS shelf off the PRE. (d) The biochemical-induced changes in DIC ($\Delta DIC^{bc}$, μmol kg$^{-1}$) vs. DO ($\Delta DO^{bc}$, μmol kg$^{-1}$) in bottom waters with depths > 10 m from Leg 2 to Leg 3. The black line denotes the slope of $\Delta DIC^{bc}$ plotted against $\Delta DO^{bc}$ derived from the Model II regression.

L439-441: Is there a reason? it would be interesting to mention it

**[Response]**: As mentioned, "sediment oxygen demand might be significant near the seabed or in its overlying water column in shallow waters (Kemp et al., 1992; Zhang and Li, 2010)", while our sampling at the bottom layer was 4-6 m above the seabed and the water depth of the hypoxic centres was nearly 20 m. We have revised the statement as "Sediment oxygen demand might be significant near the seabed or in its overlying water column in shallow waters (Kemp et al., 1992; Zhang and Li, 2010). However, our sampling at the bottom layer was 4-6 m above the seabed and the water depth of the hypoxic zones was nearly 20 m (Fig. 4). In such sampling area, Cui et al. (2019) found that oxygen losses by sediment oxygen demand (i.e., benthic respiration) were much smaller than the bacterial respiration in the water column based on both incubation experiments and oxygen budget analysis." (Page 18, Lines 415-420 of our revised MS).

L457: See previous comments about intrusions

**[Response]**: As reponsed above, deep shelf waters were actually advected onshore in the area with depths > 20 m, as the bottom waters originate from the coastal current over the

middle/outer shelf which flows northeastwardly in summer. We have replaced "shelf benthic waters" by "deep shelf waters".

L444-462: This paragraph is not very clear and I am not sure that it adds anything to your story.

**[Response]**: This paragraph explains how our estimated OCR differs from that derived from the steady-state budget analysis and how oxygen sources/sinks shift with oxygen decline for hypoxia formation. We have revised this paragraph for clearly elaborating these issues (Pages 18-19, Lines 423-442 of our revised MS).

L472-473/480-481: The hypoxia formation time scale (6-12days) is shorter than the estimate of residence time. Also 15 days in Li et al (2020) is a depth-averaged estimate so residence time in bottom waters could be significantly longer.

**[Response]**: We appreciate the reviewer's comment. We have revised the statement accordingly as "These estimates of hypoxia formation timescale are shorter than the bottom water residence time (> 15 days, Li et al. (2020)), which favoured the hypoxia formation to the west off the PRE (Su et al., 2017; Zhao et al., 2020)" (Page 19, Lines 452-453 of our revised MS) and "…extend to a larger hypoxic zone within the water residence time of > 15 days" (Page 19, Lines 460-461 of our revised MS).

L475: "shoreward-intruded": same comment as before, "shelf bottom waters" is more appropriate

**[Response]**: As reponsed above, deep shelf waters were actually advected onshore in the area with depths > 20 m, as the bottom waters originate from the coastal current over the middle/outer shelf which flows northeastward in summer. We have replaced "shelf benthic waters" by "deep shelf waters".

L498-501: rephrase for clarity.

what does that mean: "had the potential to overwhelmingly destroy the stability of the water column"? How is this defined?

**[Response]**: As mentioned in the original MS, "at least four named tropical cyclones impacted the study area from May to August in 2018, most of which … and increased the wind speed up to over 9 m s$^{-1}$, being able to destroy water column stratification and interrupt hypoxia formation (Geng et al., 2019)", which help understand these follow-up sentences. Annually, about five of the six annual tropical cyclones that travel across the northern South China Sea

from May to September (when seasonal hypoxia develops), on average, had the maximum wind speed of > 9 m s$^{-1}$ (Figure R7), which could well vertically mix the water column.

[Figure]

**Figure R7:** Statistics of tropical cyclones passing the northern South China Sea (NSCS) from May to September over 1975-2019. (a) Numbers of tropical cyclones. TD, TS, STS, TY and STY represent tropical depressions (the maximum wind speed near the centre is between 10.8-17.1 m s$^{-1}$ over its lifetime), tropical storms (17.2-24.4 m s$^{-1}$), strong tropical storms (24.5-32.6 m s$^{-1}$), typhoons (32.7-41.4 m s$^{-1}$) and strong typhoons (41.5-50.9 m s$^{-1}$), respectively. (b) The maximum wind speed of each tropical cyclone. The black line and grey shadow denote the annual average and range of the maximum wind speeds. (c) The time interval between two successive tropical cyclones. The black line and grey shadow denote the annual average and range of the time intervals. (d) The wind rose of the intensity of tropical cyclones. (e) The wind rose of the duration of tropical cyclones. The wind speed in (b) and wind direction in (d, e) were recorded at the Waglan Island station.

L508-509: This sentence is odd.

**[Response]**: As the time interval between two successive tropical cyclones was mostly less than 15 days, close to the timescale for hypoxia formation. The number of the time interval < 15 days showed a decreasing trend over 1975-2019 (Fig. R7c). Specifically, this number was averaged 4.1 per year before the year of 2000, in contrast to 2.8 per year after 2000. Therefore, hypoxia was more probably destroyed by tropical cyclones before the year of 2000, resulting in few observations of hypoxia in this study area. We have added the statements "The number of the time interval < 15 days showed a decreasing trend over 1975-2019 (Fig. 9c). Specifically, this number was averaged 4.1 per year before the year of 2000, in contrast to 2.8 per year after 2000." (Page 20, Lines 488-489 of our revised MS) to help clarify why less hypoxia was formed and even observed before the year of 2000 (Yin et al., 2004; Rabouille et al., 2008).

L511: suggestion: "Although strong winds constrain the development of hypoxia, tropical cyclones potentially..."

**[Response]**: Accepted. We have revised the sentence accordingly (Page 20, Line 494 of our revised MS).

L515-516: The reference is in chinese and therefore not accessible to most reader, please remove. You can provide a time series of rainfall in the supporting material if necessary or add it to Figure 1. However, what you mention here is river discharge and it would be more appropriate to show a time series of freshwater discharge in the PRE. This would fit well in Figure 1.

**[Response]**: We appreciate the reviewer's comment and have removed the reference. A time series of freshwater discharge has been shown in Li et al. (2021).

L518: suggested change: "... PRE did not increase significantly". See comment above about the time series of river discharge.

**[Response]**: Accepted. We have revised it accordingly (Page 20, Line 499 of our revised MS).

L521-522: remove "as shown in Fig. 3 that strong blooms occurred in the surface plume along the coast with much higher Chl *a* concentrations during Leg 2 than that during Leg 1."

**[Response]**: Here our observations are to support the statement "Enhanced vertical mixing and/or freshwater discharge supplied large amounts of nutrients to the surface layer to fuel

phytoplankton blooms following large storms (Zhao et al., 2009; Ni et al., 2016; Wang et al., 2017)". We would like to keep them.

L526: Figure 4 show that $O_2$ is decreasing in bottom waters but hypoxia did not re-establish yet.

[Response]: Hypoxia was actually re-established durng Leg 3 with DO concentrations of ~45 μmol kg$^{-1}$. It was not clearly shown in Figure 4 because of the limited hypoxia area. We have revised the statement as "Off the PRE, we also found that hypoxia re-occurred in the wake of a more extensive freshwater plume and enhanced eutrophication after the passage of typhoon SONTIHN (e.g., ~ 45 μmol kg$^{-1}$ at stations F202 and F302)" (Pages 20-21, Lines 507-509 of our revised MS).

L542: the decreasing frequency is in the tropics but not necessarily further north, e.g. a recent study with similar conclusions: https://advances.sciencemag.org/content/6/51/eabd5109

[Response]: The recent study, Chu et al. (2020), showed that "a decreased TC track density over almost the entire tropics and subtropics in response to $CO_2$ doubling. In the 4×$CO_2$ experiment, the reduction in TC density is even more pronounced, extending further into the subtropics. TC track density decreases globally by 7 and 32% in the 2×$CO_2$ and 4×$CO_2$ experiments, respectively". We have revised the statement as "In a changing climate, tropical cyclone features are believed to have a trend with higher intensity but lower frequency" (Page 21, Lines 513-514 of our revised MS).

L551: For clarity you could say that the development of hypoxia depend on a trade-off between between storm intensity and frequency.

[Response]: Accepted. We have revised the statement accordingly (Page 21, Lines 522-523 of our revised MS).

L557-568: this is more a summary than conclusions.

[Response]: We have revised the section Conclusions beyond a summary (Page 23, Lines 540-554 of our revised MS).

L564: suggested modification: "This hypoxia timescale is shorter than water residence time but comparable to the disturbance of hypoxia..."

**[Response]**: Accepted. We have revised the statement accordingly (Page 23, Line 549 of our revised MS).

L566-568: please remove

**[Response]**: We have removed this statement accordingly.

L572: http://www.geodata.cn: the website doesn't seem to be available on english. Will the data be available to the non-chinese speaking readers?

**[Response]**: Yes. We have applied for the English version and the data will later be available to non-Chinese speaking readers.

Figure 9c: why is there negative time intervals?

**[Response]**: The negative intervals indicate the partial overlaps in time of two successive tropical cyclones impacted the NSCS, that is, the second one starts to impact the NSCS before the dying-out of the first one. We have added an explanation for negative values in the figure title (Page 22, Figure 9 of our revised MS).

**Minor comments/edits:**

L191/217/244: Suggestion: add the leg # for each phase in the header, i.e. "(Leg 1)" for section 3.1

**[Response]**: We appreciate the reviewer's suggestion. Although sub-sections 3.1 and 3.3 well corresponds to Leg 1 and Leg 3, sub-section 3.2 includes both time-series observations and Leg 2. They do not follow a strict relationship. Therefore, we would like not to add the leg # in the headers.

L204 and all occurrences elsewhere: bottom seem more appropriate than benthic

**[Response]**: We appreciate the reviewer's suggestion. "Deep shelf waters" might be more appropriate than "shelf benthic waters" or "shelf bottom waters", as deep shelf waters originate from the alongshore shelf current which flows northeastwardly in summer and invade upslope off the Pearl River Estuary due to irregular topography (Gan et al., 2009; Liu et al., 2020). Therefore, we have replaced all "shelf benthic waters" by "deep shelf waters" throughout our MS.

L209: suggestion: "The general spatial pattern of hypoxic centers..."

**[Response]**: Accepted. We have revised the sentence accordingly (Page 8, Line 216 of our revised MS).

L228: suggestion: Large phytoplankton blooms were identified in the surface plume between...
**[Response]**: Accepted. We have revised the sentence accordingly (Page 9, Line 237 of our revised MS).

L235: remove "however"
**[Response]**: Accepted. We have removed "however" accordingly (Page 9, Line 244 of our revised MS).

L550: remove "enduring"
**[Response]**: Accepted. We have removed "enduring" accordingly (Page 23, Line 522 of our revised MS).

Figure 7: it should be: "Profiles of temperature (°C) (blue dashed lines)"
**[Response]**: Accepted. We have corrected the figure title as "**Figure 7**: Profiles of temperature (°C) (blue dashed lines), salinity (pink solid lines), …" (Page 15, Figure 7 of our revised MS).

Figure S2: DO: the color bar is not complete
**[Response]**: Accepted. We have corrected the color bar for DO in Figure S2 (Page 3 of our revised Supplement).

L311: (e.g., stations A8 and A11, Fig. 7a)
**[Response]**: Accepted. We have added "A11" in the sentence accordingly (Page 14, Line 322 of our revised MS).

**Anonymous Referee #2**
I thank the authors for their extensive work to revise the paper. I suggest a few final clarifications.
**[Response]**: We appreciate the reviewer's affirmation of our revisions.

(1) in the methods where the stratification calculation is described, the authors don't specifically say where and when they applied computations of stratification from their data and why. It is a small thing, but perhaps add a sentence to make clear where this stratification index was calculated.

Stratification – say where and when you calculate it (see methods)

**[Response]**: We appreciate the reviewer's suggestion. We have added the statement in the methods — "CTD temperature and salinity were used to calculate the buoyancy frequency for sampling stations, to indicate the controls of density stratification on the development of hypoxia (Section 4.1)" (Page 4, Line 116-118 of our revised MS).

(2) Line 829 – In the new discussion of tidal forcing oxygen, the authos refer to mg/L units of oxygen, which is different from the rest of the paper. Can this be referred to in umol/kg for consistency? Weight units are also used in the section beginning on 1065.

**[Response]**: We appreciate the reviewer's comment. We have unified the unit for oxygen as $\mu$ mol kg$^{-1}$ throughout our revised MS.

(3) "organic matters" should be "organic matter" throughout the manuscript.

**[Response]**: Accepted. We have replaced all "organic matters" by "organic matter" in our revised MS.

(4) Thus authors add some new discussion which says "The oxygen enriched incubation of unfiltered water samples also revealed that the OCR could be significantly enhanced when the initial *in situ* DO concentration was low (e.g., ~ 30 μmol kg$^{-1}$), but changed little when the *in situ* DO concentration was higher than ~ 90 μmol kg$^{-1}$ (He et al., 2014)." It is not entirely clear what is meant by this. Were there experiments performed where the same water sample was incubated at ambient DO (30 umol/kg; "incubation 1") and then another sample of the same water was enriched with oxygen and incubated ("incubation 2)? It seems so because the context of the statement seems to support the idea. If I am right, the sentence should be re-worded, or perhaps slightly expanded to be clear how the "enhancement" was determined.

**[Response]**: This new discussion indicates the OCR varies with the *in situ* DO concentration, that is, when *in situ* DO concentration was higher than ~ 90 μmol kg$^{-1}$, the OCR changed little, while when the *in situ* DO concentration decreased to ~ 30 μmol kg$^{-1}$, the OCR also decreased. The reviewer was right that He et al. (2014) performed the oxygen enriched incubation using

two samples of the same water: one with in situ oxygen and another one bubbled with fresh air to enrich the oxygen before incubation. We have revised the statement as "The oxygen enriched incubation of unfiltered water samples also revealed that the OCR changed little when the in situ DO concentration was higher than ~ 90 μmol kg$^{-1}$, but decreased when the in situ DO concentration decreased to ~ 30 μmol kg$^{-1}$ (He et al., 2014)." (Page 18, Line 429-432 of our revised MS).

**References**

Allahdadi, M. N. and Li, C.: Numerical Experiment of Stratification Induced by Diurnal Solar Heating Over the Louisiana Shelf, in: Modeling Coastal Hypoxia: Numerical Simulations of Patterns, Controls and Effects of Dissolved Oxygen Dynamics, edited by: Justic, D., Rose, K. A., Hetland, R. D. and Fennel, K., Springer International Publishing, Cham, 1-22, https://doi.org/10.1007/978-3-319-54571-4_1, 2017.

Chu, J.-E., Lee, S.-S., Timmermann, A., Wengel, C., Stuecker, M. F. and Yamaguchi, R.: Reduced tropical cyclone densities and ocean effects due to anthropogenic greenhouse warming, Sci Adv, 6, eabd5109, https://doi.org/10.1126/sciadv.abd5109, 2020.

Cui, Y. S., Wu, J. X., Ren, J. and Xu, J.: Physical dynamics structures and oxygen budget of summer hypoxia in the Pearl River Estuary, Limnol Oceanogr, 64, 131-148, https://doi.org/10.1002/lno.11025, 2019.

Gan, J., Cheung, A., Guo, X. and Li, L.: Intensified upwelling over a widened shelf in the northeastern South China Sea, Journal of Geophysical Research: Oceans, 114, https://doi.org/10.1029/2007JC004660, 2009.

Geng, B. X., Xiu, P., Shu, C., Zhang, W. Z., Chai, F., Li, S. Y. and Wang, D. X.: Evaluating the Roles of Wind- and Buoyancy Flux-Induced Mixing on Phytoplankton Dynamics in the Northern and Central South China Sea, J Geophys Res-Oceans, 124, 680-702, https://doi.org/10.1029/2018jc014170, 2019.

He, B. Y., Dai, M. H., Zhai, W. D., Guo, X. H. and Wang, L. F.: Hypoxia in the upper reaches of the Pearl River Estuary and its maintenance mechanisms: A synthesis based on multiple year observations during 2000-2008, Mar Chem, 167, 13-24, https://doi.org/10.1016/j.marchem.2014.07.003, 2014.

Huang, J., Hu, J., Li, S., Wang, B., Xu, Y., Liang, B. and Liu, D.: Effects of Physical Forcing on Summertime Hypoxia and Oxygen Dynamics in the Pearl River Estuary, Water-Sui, 11, https://doi.org/10.3390/w11102080, 2019.

Kemp, W. M., Sampou, P. A., Garber, J., Tuttle, J. and Boynton, W. R.: Seasonal depletion of oxygen from bottom waters of Chesapeake Bay: roles of benthic and planktonic respiration and physical exchange processes, Mar Ecol Prog Ser, 85, 137-152, http://www.jstor.org/stable/24829928, 1992.

Li, D., Gan, J., Hui, C., Yu, L., Liu, Z., Lu, Z., Kao, S.-J. and Dai, M.: Spatiotemporal Development and Dissipation of Hypoxia Induced by Variable Wind-Driven Shelf Circulation off the Pearl River Estuary: Observational and Modeling Studies, Journal of Geophysical Research: Oceans, 126, e2020JC016700, https://doi.org/10.1029/2020JC016700, 2021.

Li, D., Gan, J., Hui, R., Liu, Z., Yu, L., Lu, Z. and Dai, M.: Vortex and Biogeochemical Dynamics for the Hypoxia Formation Within the Coastal Transition Zone off the Pearl River Estuary, Journal of Geophysical Research: Oceans, 125, e2020JC016178, https://doi.org/10.1029/2020JC016178, 2020.

Liu, Z. and Gan, J.: A modeling study of estuarine–shelf circulation using a composite tidal and subtidal open boundary condition, Ocean Model, 147, 101563, https://doi.org/10.1016/j.ocemod.2019.101563, 2020.

Liu, Z., Zu, T. and Gan, J.: Dynamics of cross-shelf water exchanges off Pearl River Estuary in summer, Prog Oceanogr, 189, 102465, https://doi.org/10.1016/j.pocean.2020.102465, 2020.

Lu, Z. M. and Gan, J. P.: Controls of seasonal variability of phytoplankton blooms in the Pearl River Estuary, Deep-Sea Res Pt Ii, 117, 86-96, https://doi.org/10.1016/j.dsr2.2013.12.011, 2015.

Ni, X., Huang, D., Zeng, D., Zhang, T., Li, H. and Chen, J.: The impact of wind mixing on the variation of bottom dissolved oxygen off the Changjiang Estuary during summer, J Marine Syst, 154, 122-130, https://doi.org/10.1016/j.jmarsys.2014.11.010, 2016.

Rabouille, C., Conley, D. J., Dai, M. H., Cai, W. J., Chen, C. T. A., Lansard, B., Green, R., Yin, K., Harrison, P. J., Dagg, M. and Mckee, B.: Comparison of hypoxia among four river-dominated ocean margins: The Changjiang (Yangtze), Mississippi, Pearl, and Rhône rivers, Cont Shelf Res, 28, 1527-1537, https://doi.org/10.1016/j.csr.2008.01.020, 2008.

Su, J. Z., Dai, M. H., He, B. Y., Wang, L. F., Gan, J. P., Guo, X. H., Zhao, H. D. and Yu, F. L.: Tracing the origin of the oxygen-consuming organic matter in the hypoxic zone in a large eutrophic estuary: the lower reach of the Pearl River Estuary, China, Biogeosciences, 14, 4085-4099, https://doi.org/10.5194/bg-14-4085-2017, 2017.

Wang, B., Hu, J. T., Li, S. Y. and Liu, D. H.: A numerical analysis of biogeochemical controls with physical modulation on hypoxia during summer in the Pearl River estuary, Biogeosciences, 14, 2979-2999, https://doi.org/10.5194/bg-14-2979-2017, 2017.

Wang, D. X., Shu, Y. Q., Xue, H. J., Hu, J. Y., Chen, J., Zhuang, W., Zu, T. T. and Xu, J. D.: Relative contributions of local wind and topography to the coastal upwelling intensity in the northern South China Sea, J Geophys Res-Oceans, 119, 2550-2567, https://doi.org/10.1002/2013jc009172, 2014.

Wong, G. T. F., Ku, T.-L., Mulholland, M., Tseng, C.-M. and Wang, D.-P.: The SouthEast Asian Time-series Study (SEATS) and the biogeochemistry of the South China Sea—An overview, Deep Sea Research Part II: Topical Studies in Oceanography, 54, 1434-1447, https://doi.org/10.1016/j.dsr2.2007.05.012, 2007.

Yin, K. D., Lin, Z. F. and Ke, Z. Y.: Temporal and spatial distribution of dissolved oxygen in the Pearl River Estuary and adjacent coastal waters, Cont Shelf Res, 24, 1935-1948, https://doi.org/10.1016/j.csr.2004.06.017, 2004.

Zhang, H. and Li, S. Y.: Effects of physical and biochemical processes on the dissolved oxygen budget for the Pearl River Estuary during summer, J Marine Syst, 79, 65-88, https://doi.org/10.1016/j.jmarsys.2009.07.002, 2010.

Zhao, H., Tang, D. L. and Wang, D. X.: Phytoplankton blooms near the Pearl River Estuary induced by Typhoon Nuri, J Geophys Res-Oceans, 114, https://doi.org/10.1029/2009jc005384, 2009.

Zhao, Y., Liu, J., Uthaipan, K., Song, X., Xu, Y., He, B., Liu, H., Gan, J. and Dai, M.: Dynamics of inorganic carbon and pH in a large subtropical continental shelf system: Interaction between eutrophication, hypoxia, and ocean acidification, Limnol Oceanogr, 65, 1359-1379, https://doi.org/10.1002/lno.11393, 2020.

---

## Author Response (AR3)

*Re: Destruction and reinstatement of coastal hypoxia in the South China Sea off the Pearl River Estuary" by Yangyang Zhao et al.*

March 16, 2021

Dear Editor,

Thank you for your time in handling our paper. Enclosed please find our further revised MS entitled "Destruction and reinstatement of coastal hypoxia in the South China Sea off the Pearl River Estuary" by Yangyang Zhao et al.

We have fully considered your suggestions and revised the manuscript accordingly. We also have added the link to our data which is available in English.

Finally, we would like to take this opportunity to thank you and the reviewers for the comments and suggestions, which significantly improved the quality of the paper. We sincerely hope that our revision will meet the standards of *Biogeosciences*.

Sincerely,

Minhan Dai
Corresponding author
State Key Laboratory of Marine Environmental Science
Xiamen University
Xiamen 361005, China
Phone: 86-592-218-2132
Fax: 86-592-218-0655
E-mail: mdai@xmu.edu.cn

**Reviewer #1** (line numbers refer to the previous version)

L212-213: Please add information on the interpolation method in the manuscript

**[Response]**: We have added the interpolation method as "Within the surveyed region where DO concentrations were interpolated into 0.5'×0.5' grids using the Kriging interpolation method, …" (Page 8, Lines 219-220 of our revised MS).

L265. Please add the information on the depths in the manuscript

**[Response]**: We have added the depths of the middle layer as "Similar to Leg 1 and Leg 2, the surface waters penetrated into the subsurface layer along the coast (Fig. 4f), likely forced by the downwelling-favourable winds (Fig. 1d) (Huang et al., 2019; Li et al., 2021), augmenting temperature and DO concentrations but bringing down salinity, particularly in the mid-depth layer from 3-5 m at nearshoremost stations to ~ 15 m at offshoremost stations (Fig. S2)" (Page 12, Lines 279-280 of our revised MS).

L337. Please clarify in the manuscript what is meant by "limited spatial extent beneath the surface plume"

**[Response]**: We have revised the statement as "… resulting in an offshore or westward shift of the hypoxic zones that only partly overlapped with the surface plume in terms of their localities" (Page 14, Lines 345-346 of our revised MS).

L498-501. Please rephrase the section "the potential to overwhelmingly destroy the stability of the water column".

**[Response]**: We have revised the statement as "on average, five out of six tropical cyclones on an annual basis had maximum wind speeds exceeding 9 m s$^{-1}$, which could easily destroy the stability of the water column and replenish oxygen into the bottom waters." (Page 20, Lines 482-483 of our revised MS).

**Other comment:**

L282. Section heading. Suggested change: "biogeochemical" instead of "biochemical"

**[Response]**: Accepted. We have revised the section heading accordingly (Page 13, Line 294 of our revised MS).